

# Methane fluxes from arctic & boreal North America: Comparisons between process-based estimates and atmospheric observations

Hanyu Liu[1], Felix R. Vogel[5], Misa Ishizawa[5], Zhen Zhang[4], Benjamin Poulter[7], Doug E.J. Worthy[5], Leyang Feng[1], Anna Laure Gagné-Landmann[3], Ao Chen[1], Ziting Huang[1], Dylan C. Gaeta[1], Joe R. Melton[6], Douglas Chan[5], Vineet Yadav[2], Deborah Huntzinger[3], and Scot M. Miller[1]

[1]Department of Environmental Health and Engineering, Johns Hopkins University, Baltimore, MD, USA
[2]Jet Propulsion Laboratory, California Institute of Technology, Pasadena, CA, USA
[3]School of Earth and Sustainability, Northern Arizona University, Flagstaff, AZ, USA
[4]Institute of Tibetan Plateau Research, Chinese Academy of Sciences, Beijing, China
[5]Environment and Climate Change Canada, Toronto, ON, Canada
[6]Environment and Climate Change Canada, Victoria, BC, Canada
[7]National Aeronautics and Space Administration, Greenbelt, MD, USA

**Correspondence:** Hanyu Liu (hliu154@jhu.edu)

**Abstract.** Methane ($CH_4$) flux estimates from high-latitude North American wetlands remain highly uncertain in magnitude, seasonality, and spatial distribution. In this study, we evaluate a decade (2007 – 2017) of $CH_4$ flux estimates by comparing 16 process-based models with atmospheric $CH_4$ observations collected from in situ towers. We compare the Global Carbon Project (GCP) process-based models with a model inter-comparison from a decade earlier called The Wetland and Wetland

$CH_4$ Intercomparison of Models Project (WETCHIMP). Our analysis reveals that the GCP models have a much smaller inter-model uncertainty and have an average magnitude that is a factor of 1.5 smaller across Canada and Alaska. However, current GCP models likely overestimate wetland fluxes by a factor of two or more across Canada and Alaska based on tower-based atmospheric $CH_4$ observations. The differences in flux magnitudes among GCP models are more likely driven by uncertainties in the amount of soil carbon or spatial extent of inundation than in temperature relationships, such as $Q_{10}$ factors. The GCP

models do not agree on the timing and amplitude of the seasonal cycle, and we find that models with a seasonal peak in July and August show the best agreement with atmospheric observations. Models that exhibit the best fit to atmospheric observation also have a similar spatial distribution; these models concentrate fluxes near Canada's Hudson Bay Lowlands (HBL). Current, state-of-the-art process-based models are much more consistent with atmospheric observations than models from a decade ago, but our analysis shows that there are still numerous opportunities for improvement.





## 1 Introduction

Natural sources of CH$_4$ contribute ∼40% of total global fluxes, and wetlands are possibly the largest single source (e.g., Kirschke et al., 2013; Saunois et al., 2020). Understanding the magnitude, seasonality, and spatial distribution of wetland CH$_4$ fluxes is important to accurately predicting future carbon-climate feedbacks. However, the response of wetland CH$_4$ fluxes to temperature changes is uncertain (e.g., Zhang et al., 2023, 2017), especially in high-latitude regions where warming occurs 2-4

times faster than the global average (e.g., Rantanen et al., 2022).

At least some of this uncertainty is related to uncertain permafrost dynamics. Permafrost covers approximately ∼15% of the land in the Northern Hemisphere (Obu, 2021), and it serves as a massive reservoir for carbon. Globally, permafrost regions store about 1,000 to 1,672 peta-grams (Pg) of soil organic carbon (SOC), nearly twice the total amount of carbon in the atmosphere (Schuur et al., 2015; Hugelius et al., 2014; van Huissteden and Dolman, 2012). As permafrost thaws, it changes the soil

environment and triggers microbial decomposition of the stored organic matter. When the soil is wet, microbial decomposition in permafrost leads to the release of CH$_4$ through the process of anaerobic respiration. One study indicates that wetland CH$_4$ fluxes can be large enough to flip some high latitude regions from a net carbon sink to a net source (Watts et al., 2023).

To understand high-latitude wetland CH$_4$ fluxes and better predict future warming, process-based (bottom-up) models are important as they can be used to estimate current wetland CH$_4$ fluxes and provide insights to future CH$_4$ projections from

regional to global scales, leveraging current scientific knowledge of different biogeochemical processes (e.g., Saunois et al., 2024; Nzotungicimpaye et al., 2021; Melton et al., 2013; Zhang et al., 2017). Despite their importance, the CH$_4$ flux estimates from bottom-up models can have large discrepancies and uncertainties. For example, bottom-up estimates show that total global wetland fluxes range from 100 to 256 Tg CH$_4$ yr$^{-1}$ (Xiao et al., 2024; Zhang et al., 2024; Saunois et al., 2020; Liu et al., 2020). In boreal North America, process-based models also estimate wetland CH$_4$ fluxes ranging from 13.8 to 39.6 Tg CH$_4$ per year

(Poulter et al., 2017). In addition, a recent study suggests an increase of 50 to 150% in global wetland CH$_4$ fluxes by 2100, a large range of numbers which points to large uncertainties in current projections (Koffi et al., 2020). Model inter-comparison projects like the Wetland and Wetland CH$_4$ Intercomparison of Models Project (WETCHIMP) have been used to compare the state-of-the-art wetland CH$_4$ flux models across different regions of the globe (e.g., Miller et al., 2016b; Wania et al., 2013; Miller et al., 2016a; Bohn et al., 2015). In more recent years, the Global Carbon Project (GCP) has been created to synthesize

scientific knowledge of the global carbon cycle, and this effort includes a large ensemble of the latest process-based CH$_4$ flux models (Poulter et al., 2017; Zhang et al., 2024). Projects like WETCHIMP and GCP make it easier to identify and diagnose uncertainties in wetland flux models because all modeling groups use similar modeling protocols, meteorological inputs, and, in some cases, common inundation or wetland maps. However, there is limited knowledge on how these models have improved or evolved over time compared to the earlier WETCHIMP inter-comparison.

A handful of studies have used approaches such as atmospheric modeling and inverse modeling to suggest improvements to process-based CH$_4$ flux models across high latitudes (e.g., Miller et al., 2016a; Karion et al., 2016). For example, several existing studies have used intensive aircraft campaign data to quantify CH$_4$ fluxes from Alaska and provide a range of estimates from 1.48 Tg CH$_4$ per year to 2.9 Tg CH$_4$ per year (Miller et al., 2016b; Hartery et al., 2018; Chang et al., 2014; Sweeney et al.,



2022). Other studies focused on $CH_4$ fluxes from high latitude North America use in situ $CH_4$ observations from long-term tower observation sites, and these studies provide a range of flux estimates from 14.8 to 19.5 Tg $CH_4$ per year for Canada and 1.56 to 3.4 Tg $CH_4$ per year for the Hudson Bay Lowlands (HBL), a prominent wetland region in northern Canada (e.g., Ishizawa et al., 2024; Miller et al., 2014; Pickett-Heaps et al., 2011; Thompson et al., 2017). Although top-down studies provide relatively good agreement on flux totals from these regions, it is also worth noting that top-down studies do not always agree, particularly on topics like seasonality and inter-annual variability. For example, Pickett-Heaps et al. (2011) indicates that there is a sharp decrease in $CH_4$ fluxes in September across the Hudson Bay Lowlands (HBL), while Thompson et al. (2017) argues that maximum $CH_4$ fluxes occur in August and September. In addition, Sweeney et al. (2016) argues that there is no multi-decadal trend in $CH_4$ fluxes using observations from Utqiagvik, Alaska, while inverse modeling studies by Thompson et al. (2017) and Ishizawa et al. (2024) identify significant inter-annual variability in fluxes across high-latitude North America. The coarse spatial resolution of some inverse estimates can further limit comparison with process-based flux models. These limitations and disagreements notwithstanding, results from top-down studies often provide better constraints on $CH_4$ fluxes over large regional domains with a narrower range of uncertainties compared to process-based models.

In this study, we use atmospheric $CH_4$ observations from tower sites to evaluate the GCP process-based models across high-latitude North America. We specifically use four sets of analyses to compare atmospheric $CH_4$ observations and the GCP wetland flux models with a goal of suggesting future improvements to these models. For each of these analyses, we run each GCP flux estimate through an atmospheric transport model to simulate atmospheric $CH_4$, and we compare the results against available atmospheric $CH_4$ observations. First, we compare the GCP models across high latitudes against the WETCHIMP models and explore how process-based flux models have evolved over the past decade. Several existing studies have evaluated the WETCHIMP models using atmospheric observations, and this retrospective comparison provides useful context on how the state of science has changed since those studies (e.g., Miller et al., 2016b; Wania et al., 2013; Miller et al., 2016a; Bohn et al., 2015). Second, we examine how the GCP models vary in $CH_4$ flux magnitude and what potential factors might drive agreement or disagreement among the models. Third, we investigate differences in seasonal cycles across models that best match atmospheric observations versus models that show seasonal discrepancies with atmospheric observations. Lastly, we examine the spatial distribution of the $CH_4$ fluxes estimated by the GCP models and identify spatial patterns that appear to yield better agreement with available atmospheric $CH_4$ data. We note that the GCP models are global in scale and not specifically designed for high-latitude regions. With that said, state-of-the-art process-based models should ideally provide accurate flux estimates across all global regions, and we argue that regional comparisons are important to inform future model development.

## 2 Data and Methods

To better understand current wetland $CH_4$ fluxes, we compare GCP $CH_4$ flux estimates with 11 years of in situ tall tower data from the United States and Canada, spanning 2007 to 2017. We focus on the months of May through October each year. Wetland $CH_4$ fluxes are largest during these months, and many existing top-down studies have focused on these months for



their analyses (e.g., Miller et al., 2016b; Chang et al., 2014; Pickett-Heaps et al., 2011). By contrast, the ratio of wetland fluxes to anthropogenic $CH_4$ emissions is much smaller in other months of the year across Alaska and Canada, making it more difficult to uniquely constrain wetland fluxes using atmospheric observations. The geographic domain of this study covers the high-latitude regions of North America, ranging from 40° N to 80° N and 170° W to 50° W.

## 2.1 Atmospheric $CH_4$ Measurements

In this study, we use continuous atmospheric $CH_4$ measurements from in situ towers across the Canada and the US between years 2007 and 2017, and the atmospheric data come from the NOAA Observation Package (ObsPack) $CH_4$ GlobalViewPlus v5.1 dataset (Di Sarra et al., 2023). There are 21 available tower sites within the study domain, and the towers provide a combination of continuous and flask measurements. We list a more detailed description of each tower site and its location in Table S1. We extract afternoon averages of the observations between 1pm and 6pm local time when the boundary layer is generally well-mixed, an approach similar to multiple existing top-down studies (e.g., Miller et al., 2014, 2016a; Karion et al., 2016; Ishizawa et al., 2024). During this time of day, $CH_4$ measurements are arguably influenced by fluxes from a broader region than at night. By contrast, the atmosphere is usually stable in the morning and at night with lower boundary layer heights, making accurate atmospheric trace gas modeling challenging.

We note that several previous studies have already used intensive aircraft campaigns to examine regional $CH_4$ fluxes across high-latitude North America (e.g., Miller et al., 2016b; Hartery et al., 2018; Chang et al., 2014; Sweeney et al., 2022). These studies use inverse modeling to provide detailed evaluations of several key aspects of $CH_4$ fluxes including magnitude and seasonality. Existing, intensive aircraft campaigns are largely centered in Alaska, while tower-based measurements offer broad spatial coverage across North America. In the present study, we focus on tower data to evaluate $CH_4$ dynamics broadly across northern North America, and we refer the reader to the aforementioned studies for detailed analyses of intensive aircraft data.

## 2.2 Global Carbon Project Models (GCP Models)

The GCP includes global-scale wetland $CH_4$ flux models that use diverse hydrological and biogeochemical schemes. The most recent GCP model ensemble includes 16 process-based models spanning the period from 2000 to 2020, though some models end earlier or later than 2020. A general description of these GCP models is provided in Table S2 and in Zhang et al. (2024). Each of the these models is run in two different ways: diagnostically and prognostically. The diagnostic runs from each model are constrained by a predefined inundation map from the Wetland Area and Dynamics for Methane Modeling version 2 (WAD2Mv2) product, while each modeling group can determine their own inundation map for the prognostic runs (Zhang et al., 2021). Note that each modeling group submitted estimates of $CH_4$ fluxes to the GCP, but the submissions do not include variables like soil carbon. This fact limits our ability to diagnose disagreements in the $CH_4$ flux estimates from different models. More detailed descriptions of the current GCP model ensemble, including their approaches to wetland inundation and model parametrization can be found in Zhang et al. (2024).

In this study, we evaluate the 11 prognostic and 16 diagnostic models included in the GCP ensemble. Each of these models was run using two different meteorological reanalysis products to examine the effects of meteorological uncertainties on esti-



mated $CH_4$ fluxes. These products include the Global Soil Wetness Project Phase 3 (GSWP3) and the Climate Research Unit Time-Series 4.06 (Harris et al., 2022; Lange and Büchner, 2020). A recent study showed that the differences between these two climate-forcing datasets are negligible (Ito et al., 2023). Nevertheless, both datasets are included in this study to provide a comprehensive evaluation. We regrid these GCP models into an uniform spatial resolution of 1° latitude by 1° longitude. This regridding process is performed using the "remapcon" function from the Climate Data Operators (CDO) software, which

conserves the total fluxes of each model during interpolation (Schulzweida, 2023).

## 2.3 Anthropogenic $CH_4$ Emissions

Anthropogenic $CH_4$ fluxes are not clearly known and are often underestimated in high-latitude North American regions, including Alberta and other parts of Canada. For example, several existing studies estimate Canadian anthropogenic fluxes ranging from 3.7 to 6.1 Tg of $CH_4$ per year (Thompson et al., 2017; Scarpelli et al., 2021; Lu et al., 2022). Baray et al. (2021) also

suggest that $CH_4$ fluxes from the Canadian energy and agriculture sectors are likely ∼59% higher than those reported in the national inventory. As a result, we include three distinct combinations of anthropogenic $CH_4$ flux products to highlight the variability and uncertainty in our analysis due to anthropogenic $CH_4$ fluxes. This approach allows us to capture a range of anthropogenic $CH_4$ flux estimates, which helps us to better understand the uncertainties associated with human-related $CH_4$ fluxes and their potential impact on high-latitude North America regions.

We use three specific anthropogenic flux products and regrid them to a spatial resolution of 1° latitude by 1° longitude for the study domain:

1. CarbonTracker $CH_4$ 2023 (Oh et al., 2023).

2. A combination of the gridded U.S. Greenhouse Gas Inventory (Version 2), and a gridded inventory of Canada's anthropogenic $CH_4$ fluxes (Monforti Ferrario et al., 2021; Maasakkers et al., 2023; Scarpelli et al., 2021).

3. The Copernicus Atmosphere Monitoring Service (CAMS) (Granier et al., 2019).

CarbonTracker is a data assimilation system designed to estimate $CH_4$ fluxes on a global scale (Oh et al., 2023). Scarpelli et al. (2021) constructed a Canadian anthropogenic flux inventory based on the Canadian National Inventory Report (NIR), the Canadian Greenhouse Gas Reporting Program (GHGRP), and other datasets to provide a detailed sectoral breakdown of fluxes. Meanwhile, Maasakkers et al. (2023) created a U.S. gridded inventory integrating data from the U.S. Environmental

Protection Agency's (EPA) Greenhouse Gas Inventory (GHGI) to provide fluxes from different sectors. CAMS is a global data assimilation system that provides estimates of global atmospheric $CH_4$ fluxes and atmospheric mixing ratios. This product is derived from a combination of the EDGARv4.3.2 and Community Emissions Data System (CEDSv3) inventories, and the product includes estimates of fluxes from different source sectors (Granier et al., 2019).





## 2.4  Atmospheric Modeling Framework

We simulate the atmospheric transport of $CH_4$ and $CH_4$ fluxes using the WRF-STILT (The Weather Research and Forecasting-Stochastic Time-Inverted Lagrangian Transport) model, which has been widely used in numerous studies of regional greenhouse gas fluxes (e.g., Miller et al., 2016b; Henderson et al., 2015; McKain et al., 2015; Kort et al., 2010; Feng et al., 2023; Miller et al., 2014). STILT is a Lagrangian particle dispersion model that simulates atmospheric transport using an ensemble of tracer particles (Lin et al., 2003). For the setup here, the model releases those particles from each measurement site, and
the particles travel backward in time for 10 days following the wind fields in WRF meteorology. STILT uses these particle trajectories to calculate surface influence maps or footprints for each atmospheric $CH_4$ observation (Figure 1). These footprint maps have units of mixing ratio per unit flux (ppb per $\mu$mol m$^{-2}$ s$^{-1}$) on a 1° by 1° grid, and we can directly multiply $CH_4$ fluxes from the process-based models with these footprint maps to predict atmospheric $CH_4$ mixing ratios at each tower site. Specifically, the footprints used in this study are generated as part of the NOAA CarbonTracker-Lagrange project and are
available from 2007 to May 2018, which defines our study time frame (Hu et al., 2019).

Since $CH_4$ has an atmospheric lifetime of about 10 years, it can remain in the atmosphere and travel around the globe. To account for the large-scale movements of $CH_4$, we estimate $CH_4$ boundary conditions using $CH_4$ observations collected over the Pacific and Atlantic oceans, from high-altitude tower sites in the continental US, and from regular aircraft flights across the US and Canada. We use these observations to interpolate a curtain of $CH_4$ mixing ratios around the boundaries of the
model domain. For each STILT simulation, we sample from this boundary condition curtain based on the ending locations of the particle trajectories. This procedure thus accounts for $CH_4$ that enters the domain from other regions of the globe. The approach used here is identical to that used in numerous existing regional $CH_4$ studies (e.g., Miller et al., 2013, 2014, 2016a).

We note that the STILT particle trajectories used here from CarbonTracker-Lagrange do not include atmospheric oxidation processes. However, $CH_4$ oxidation by hydroxyl radicals likely has a small impact in our study given the short, 10-day time
frame of the regional STILT simulations used here. For example, Miller et al. (2013) argue that $CH_4$ mixing ratios decay less than 1 to 1.5 ppb during the course of a typical STILT simulation, based on an analysis of in situ observation sites in the continental US and estimated OH fields from GEOS-Chem. Overall, this decay is less than 5% of the average, total modeled $CH_4$ enhancements in this study. The impact of OH in our study may be even smaller because OH mixing ratios are usually lower at high latitudes.

We combine the aforementioned modeling components using the following equation to compare atmospheric $CH_4$ observations with the STILT model predictions using the GCP flux models:

$$\boldsymbol{Z} \sim \mathbf{H}\big[\boldsymbol{s} + \boldsymbol{A} + \boldsymbol{B}\big] + \boldsymbol{b}. \tag{1}$$

where $\boldsymbol{Z}$ represents the atmospheric observations from the in situ towers across the US and Canada (dimensions $n \times 1$, where $n$ are the number of observations). $\mathbf{H}$ is a matrix of influence footprints assembled from the WRF-STILT model, showing how
surface fluxes from different locations and times contribute to the observations (dimensions $n \times m$, where $m$ is the number of flux model grid boxes in space and time). Within the brackets, $\boldsymbol{s}$ refers to wetland $CH_4$ flux estimates from the process-based




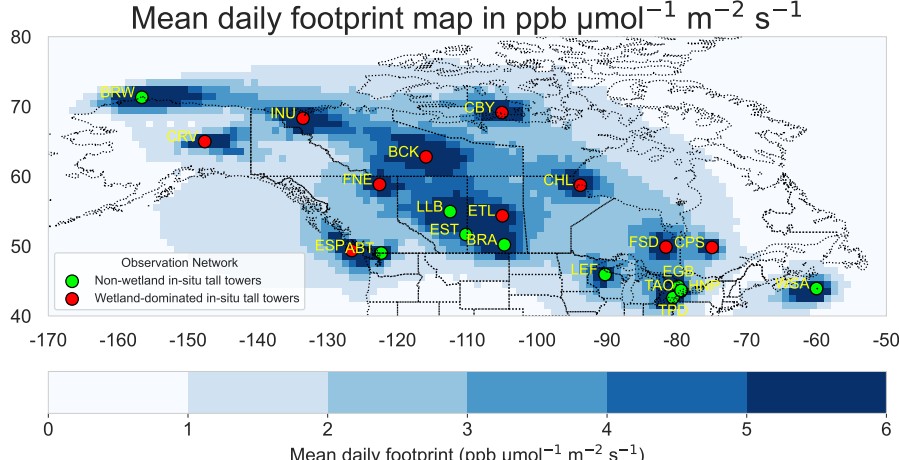

**Figure 1.** The US and Canadian atmospheric $CH_4$ observing network from 2007-2017. The figure also shows the WRF-STILT mean daily footprint map in ppb $\mu$mol$^{-1}$ m$^{-2}$ s$^{-1}$ across the study domain of 40°N to 80°N and 170°W to 50°W. Red circle dots show in situ tall tower sites from NOAA and Environmental Canada from the ObsPack GlobalViewPlus v5.1 dataset (Di Sarra et al., 2023). The lime-colored dots represent non-wetland sites, where the wetland-to-anthropogenic $CH_4$ concentration ratio is less than 1.5 (using anthropogenic emissions from the CAMS product). In contrast, the red-colored dots indicate wetland-dominated sites, where this ratio exceeds 1.5.

GCP models (dimensions $m \times 1$, Sect. 2.2), $\boldsymbol{A}$ refers to the anthropogenic $CH_4$ fluxes estimate from one anthropogenic product (dimensions $m \times 1$, Sect. 2.3), and $\boldsymbol{B}$ denotes biomass burning fluxes from the Global Fire Emissions Database (GFED v4.1) (Randerson et al., 2017) (dimensions $m \times 1$). The last variable, $\boldsymbol{b}$, represents the $CH_4$ boundary condition (dimensions $m \times 1$).

Note that we primarily analyze tower-based observations sites where the average ratio of modeled $CH_4$ from STILT using all GCP models to modeled $CH_4$ from STILT using the CAMS anthropogenic flux product is higher than 1.5 (sect.2.4, sect.2.2, sect.2.3). This screening means that the wetland contributions at each site are at least $50\%$ higher than the likely influence of anthropogenic emissions, and we exclude the other sites to prioritize wetland $CH_4$ dominated regions (see Table S3 for additional details). In this study, we also include ETL (East Trout Lake) and FNE (Fort Nelson) because their ratios are close to 1.5 and we want to include as many sites as possible to have a broader spatial coverage. We focus on these sites because we aim to better quantify the contribution of wetlands to atmospheric $CH_4$ levels while minimizing the confounding effects of anthropogenic sources, the magnitudes of which are also uncertain. The ten final sites that we include within this study are: Churchill, Manitoba (CHL), Cambridge Bay, Nunavut Territory (CBY), East Trout Lake, Saskatchewan (ETL), Estevan Point, British Columbia (ESP), Fort Nelson, British Columbia (FNE), Fraserdale, Ontario (FSD), Inuvik, Northwest Territories (INU), Behchoko, Northwest Territories (BCK), Chapais, Quebec (CPS), and the Carbon in Arctic Reservoirs Vulnerability Experiment Tower, Fairbanks (CRV) (see Table S1 for additional details). The remaining sites that are not included in this analysis are towers in urban environments (e.g., sites in the Toronto and Vancouver metropolitan areas); towers close to oil and





gas production in Alberta, Canada, or Prudhoe Bay, Alaska; towers that are frequently used as clean air background sites (e.g., Sable Island, Nova Scotia or WSA), and sites proximal to intensive agriculture.

## 2.5 Temperature Sensitivity

We assess the relationship between wetland $CH_4$ fluxes from the GCP models and temperatures by fitting $Q_{10}$ curves for each GCP model. The $Q_{10}$ factor illustrates how $CH_4$ wetland fluxes change with a per 10-degree change in temperatures, and a higher $Q_{10}$ means that wetland fluxes are more sensitive to temperature changes (e.g., Mundim et al., 2020; James, 1953; van Hulzen et al., 1999). Several of the GCP models explicitly include a $Q_{10}$ function within the model equations, whereas other models use different functions or modeling schemes to parameterize the relationships between $CH_4$ fluxes and temperature. Even though not all of the GCP models explicitly use a $Q_{10}$ function, we nevertheless fit each of the flux estimates to a $Q_{10}$ function. Doing so allows us to directly compare the apparent temperature relationships in the different GCP models. Furthermore, to account for the impact of inundation dynamics, we adjust the fluxes by multiplying them by the corresponding inundation fraction at each grid cell. This adjustment normalizes the fluxes to a standard wetland area, demonstrating a more consistent comparison of how wetland $CH_4$ fluxes respond to temperature variations.

The following formula represents the $Q_{10}$ function (e.g., Zhang et al., 2024; Mundim et al., 2020):

$$R(T) = R_b \cdot Q_{10}^{\frac{(T - T_{\text{ref}})}{10}} \tag{2}$$

where $R(T)$ are monthly wetland $CH_4$ fluxes at near-surface air temperature T (°C) based on the same meteorological products used to generate the GCP models (Sect. 2.2), and $R_b$ is the baseline flux at a reference temperature. In this study, we set the reference temperature $T_{ref}$ at 15°C, and the exponential term shows the difference between an ambient temperature and the reference temperature of 15°C, capturing the proportional change in wetland $CH_4$ flux with temperature. We use the Nelder-Mead method to simultaneously optimize the parameters $R_b$ and $Q_{10}$ by minimizing the sum of squared errors between the predicted fluxes $R(T)$ and the actual wetland $CH_4$ fluxes from the GCP models (Gao and Han, 2012).

## 3 Results and Discussions

In this section, we compare the modeled $CH_4$ mixing ratios using the GCP models to atmospheric observations. We use these comparisons to evaluate the magnitude, seasonality, and spatial distribution of the GCP flux models. In each subsection, we also speculate on the possible reasons driving the agreement or disagreements that we see in our analyses. Note that we do not include an extensive discussion of inter-annual variability (IAV) in our analysis; uncertainties in anthropogenic $CH_4$ sources lead to large uncertainties in our inferences about wetland fluxes, and we argue that it would be difficult to constrain IAV in wetland fluxes across Alaska and Canada without accurate knowledge of IAV from anthropogenic sources. Disentangling these changes from changes in $CH_4$ fluxes due to wetlands is a challenge, and existing studies reach conflicting conclusions (e.g., Ishizawa et al., 2024; Thompson et al., 2017).



## 3.1 Comparisons Between the GCP and WETCHIMP Models

The GCP model ensemble is an updated version of the earlier WETCHIMP inter-comparison over a decade ago, and these
projects share five common models (LPJ-Bern, LPJ-wsl, ORCHIDEE, SDGVM, DLEM) (Melton et al., 2013). Overall, we
find that, compared to the WETCHIMP models, the GCP models have a smaller flux magnitude, better consensus on flux
magnitude, and better agreement on the spatial distribution of fluxes within our study domain. This result points to an evolution
and growing consensus among state-of-the-art wetland $CH_4$ flux models.

We find that the $CH_4$ flux estimates from the GCP models are much smaller across most of high-latitude North America
compared to the WETCHIMP models. We calculate annual $CH_4$ flux totals for Canada using the 11 prognostic and 16 diagnos-
tic GCP models with both climate forcing datasets (GSWP3 and CRU), and the uncertainty bars in Fig. 2 represent the standard
deviation of the $CH_4$ flux estimates among models within the same group. The mean annual $CH_4$ flux total for Canada using
the 11 prognostic GCP models with CRU is $14.19 \pm 7.41$ Tg $CH_4$ per year, and the mean using the 16 diagnostic models with
CRU is $12.17 \pm 5.48$ Tg $CH_4$ per year (Figure 2). In contrast, the Canadian annual $CH_4$ flux total using the seven WETCHIMP
models with CRU is a factor of more than ~1.5 higher than the prognostic and diagnostic GCP models, with flux estimates
(based on the standard deviations of models within the same group) of $21.50 \pm 15.12$ Tg $CH_4$ per year. We notice that the annual
Canadian $CH_4$ flux total for the LPJ-WHyMe model from WETCHIMP is $46.25 \pm 5.88$ Tg $CH_4$ per year (Fig. S5). Therefore,
we subsequently exclude this model to recalculate the annual $CH_4$ flux total using the other six WETCHIMP models, and
evaluate whether or not it brings the flux estimates similar to the GCP models. However, the annual $CH_4$ flux total using the
other six WETCHIMP models with CRU is $17.97 \pm 12.59$ Tg $CH_4$ per year, which is still about a factor of ~1.4 higher than
the prognostic GCP models using CRU meteorology. In addition, the annual $CH_4$ flux totals estimated by the WETCHIMP
models are a factor of ~1.3 or higher than the GCP models in the two dominant high-latitude biomes across North America
(tundra and boreal forests) (Fig. 2). In Alaska, the annual $CH_4$ flux total estimated by the 11 prognostic GCP models with CRU
is $1.31 \pm 0.85$ Tg $CH_4$ per year, whereas the seven WETCHIMP models yield a higher value of $1.66 \pm 2.02$ Tg $CH_4$ per year.
Across the North American boreal forests and tundra, the annual $CH_4$ flux totals estimated by the 11 prognostic GCP models
with CRU are $10.71 \pm 5.73$ and $1.64 \pm 1.31$ Tg $CH_4$ per year, respectively. In comparison, the annual $CH_4$ flux totals estimated
by the seven WETCHIMP models in these two biomes are $16.62 \pm 8.55$ and $2.15 \pm 1.34$ Tg $CH_4$ per year, respectively.

We also find that the $CH_4$ fluxes estimated by the 11 prognostic GCP models result in much lower inter-model uncertainty
compared to the seven WETCHIMP models, with smaller inter-model disagreement across Canada and southern Alaska. To
evaluate model agreement on the spatial distribution of fluxes, we compare the inter-model uncertainty or the standard deviation
of flux estimates for each individual model grid box of the GCP and WETCHIMP models. Since each WETCHIMP model
identifies the inundation or wetland area differently, we compare these models with the prognostic GCP models (Melton et al.,
2013). Note, however, that not all of the WETCHIMP modeling groups generated their own wetland or inundation maps
prognostically, and some, like LPJ-Bern and LPJ-WHyMe, use a constant, prescribed wetland map. In Figure 3, darker shades
at each grid box represent higher inter-model uncertainty across these process-based models. We observe that the GCP models
have much lighter shades across the study domain, indicating better inter-model agreement.





We further find that the WETCHIMP models generally exhibit seasonal cycles that are similar to the GCP models (Figs. S1a and S1b). Most WETCHIMP models estimate peak $CH_4$ fluxes across Alaska and Canada in July and August, except CLM4Me (which peaks in June) and LPX-Bern (which peaks in September). These small model disagreements notwith-standing, this result illustrates that the seasonal cycles of the GCP models have not changed markedly from the WETCHIMP models. Inter-model agreement on the magnitude and spatial distribution of fluxes improved in the GCP ensemble compared to the WETCHIMP ensemble, but we find no such convergence in model agreement on the seasonal cycle. The WETCHIMP models already showed relatively good agreement on the seasonal cycle of fluxes, so there was not much opportunity for im-provement. Furthermore, the seasonal cycle of these model estimates is largely dependent on temperature, meaning that it is arguably easier to model than other features that depend on more complex processes.

The reduction in inter-model uncertainties from WETCHIMP to GCP may relate to how the models estimate wetland distri-bution. Different WETCHIMP model yield very different estimates of maximum wetland extent – from 2.7 to $36.4 \times 10^6$ km$^2$ for the global extra-tropics ($> 35°$N), depending upon the model. Melton et al. (2013) explain that several WETCHIMP models use a binary approach to identify wetland areas, where individual model grid boxes are either 100% wetland or 0% wetland, and these models tend to have $\sim 3-4$ times greater wetland area compared to other models (Fig.2 and Table.2 in (Melton et al., 2013)). By contrast, other WETCHMIMP models were parameterized to match remote sensing estimates of wetland or open water extent. In contrast to WETCHIMP, the GCP model ensemble also includes diagnostic experiments in which all modeling groups used the WAD2M v2 inundation map. These efforts to create a standardized, diagnostic map of wetland extent may have also influenced the prognostic GCP experiments, and modeling groups may have tuned or modified their setup to be more consistent with the diagnostic model simulations. In addition, the lower magnitude of $CH_4$ fluxes estimated by the GCP models (compared to the WETCHIMP models) is partly attributed to efforts by the GCP modeling group to reduce double-counting of freshwater areas (e.g., lakes and ponds) in WAD2M v2 (Zhang et al., 2021).

This improved inter-model agreement implies that the fluxes estimated by the current process-based GCP models are more accurate compared to the fluxes estimated by the previous WETCHIMP models, though that outcome is not guaranteed. In the following sections, we compare GCP models with atmospheric observations as a way to gauge whether the GCP models are indeed more skilled at capturing $CH_4$ fluxes across high-latitude North America.

### 3.2 Flux Magnitudes of GCP Models

We find that even though the mean wetland $CH_4$ fluxes of the GCP models are about a factor of 2 lower than the WETCHIMP models across northern North America, most of them are still likely an overestimate by a factor of 2 or more compared to atmospheric $CH_4$ observations (Fig. 4). We evaluate the magnitude of the GCP models by comparing modeled mixing ratios from STILT against observations at the tower sites. Specifically, we divide modeled $CH_4$ mixing ratios using wetland fluxes from the GCP models by the observed increments, shown in Fig. 4. The modeled wetland $CH_4$ mixing ratios are calculated by passing each of the GCP models through STILT. The observed increments are calculated as the atmospheric $CH_4$ observations minus factors unrelated to wetlands – the $CH_4$ boundary condition and the contributions of anthropogenic and biomass burning fluxes at the observation sites. In Fig. 4, we compare the magnitude of the modeled wetland $CH_4$ mixing ratios and the observed





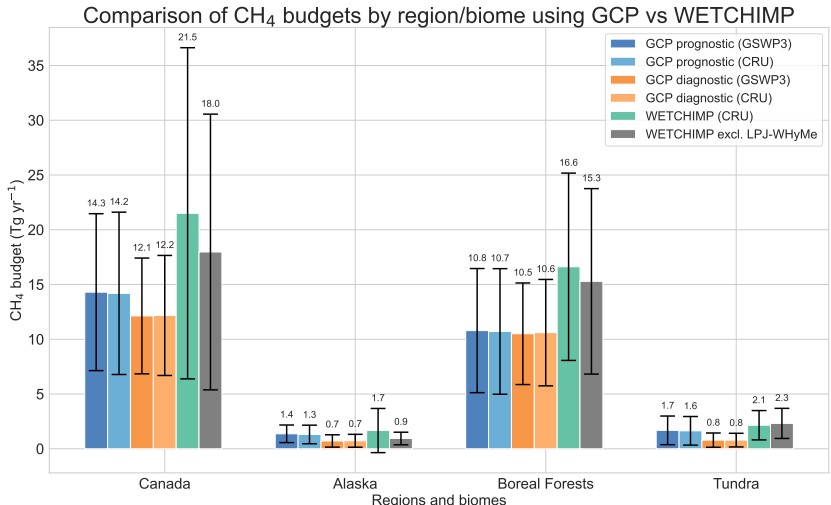

**Figure 2.** Annual $CH_4$ flux totals across Canada, Alaska, and several biomes. The four bars on the left of each region or biome represent the 2 different climate forcing data (GSWP3 and CRU) and prognostic versus diagnostic types for the GCP models. The green bar shows the mean annual $CH_4$ flux total using all WETCHIMP models, and the gray bar denotes the mean flux total excluding the LPJ-WHyMe model. The unit of the annual wetland $CH_4$ flux totals is Tg $CH_4$ per year.

increments at each wetland-dominated in situ tower site across high-latitude North America. A factor larger than one means that the mixing ratios of modeled wetland $CH_4$ using the GCP models are higher than the observed increments. By contrast, the gray dashed line at the y-axis equal to 1 indicates a perfect alignment between the modeled wetland $CH_4$ mixing and the observed increment. The error bars in Fig. 4 reflect the range of results when we use different anthropogenic flux estimates in the calculations (Sect. 2.3). Note that CH4MOD, DLEM, LPJ-GUESS, TEM-MDM, and TRIPLEX-GHG only have diagnostic simulations and not prognostic simulations, and their diagnostic comparisons are represented exclusively by orange bars.

Based on these results, we also argue that anthropogenic $CH_4$ fluxes pose an enormous challenge for isolating and quantifying $CH_4$ fluxes from wetlands, even at very remote observation sites in Canada and Alaska. The vertical bars in Fig. 4 indicate uncertainties in the results due to uncertain anthropogenic fluxes, and we observe a broad spectrum of values depending on which anthropogenic $CH_4$ flux estimate we use. For example, modeled mixing ratios from STILT using the GCP $CH_4$ model CLASSIC run prognostically are anywhere between ∼2.5 times higher than the observed increment to ∼6 times higher, depending on the choice of anthropogenic flux product. As a result, we cannot precisely constrain the optimal magnitude of wetland fluxes. These uncertainties notwithstanding, our findings still suggest that wetland fluxes estimated by the 11 prognostic and 16 diagnostic models are often higher than implied by atmospheric observations.

It is difficult to determine the specific causes that drive model disagreements over the magnitude of wetland $CH_4$ fluxes. However, these variations are more likely influenced by factors such as soil carbon or by the simplicity/complexity of the model structure rather than by disagreements over the effects of temperature on fluxes. We do not have a comprehensive set of modeled environmental variables (e.g., soil carbon) to conduct a systematic examination of all sources of uncertainty. However,





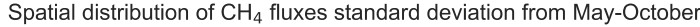

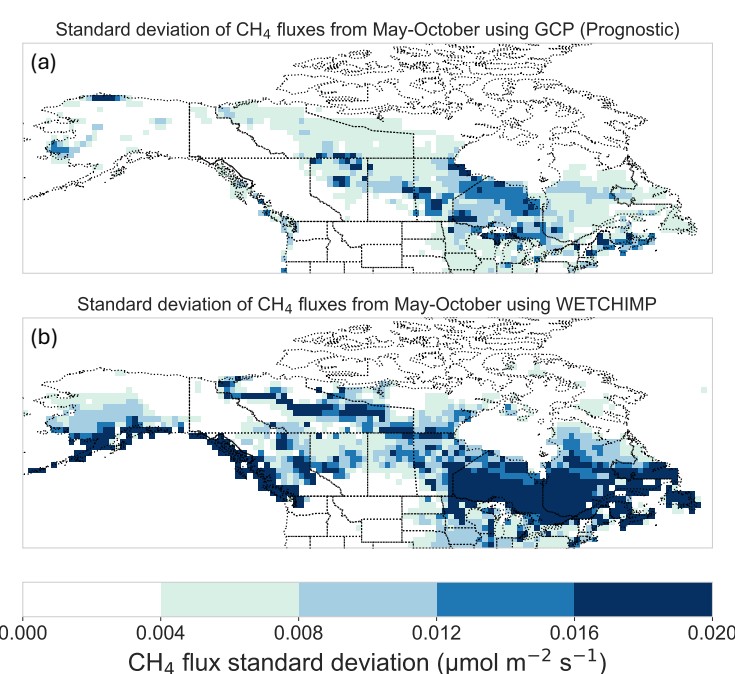

**Figure 3.** The inter-model standard deviation for each individual model grid box, calculated using the 11 prognostic GCP models (top) and WETCHIMP models (bottom). The inter-model uncertainty in mode locations is higher for the WETCHIMP models than the GCP models. All fluxes have units $\mu$mol m$^{-2}$ s$^{-1}$.

the available model outputs allow us to reason through some key contributors to these uncertainties, such the relationships
between fluxes and temperature (i.e., estimated $Q_{10}$ values) and the effects of using a common diagnostic inundation map versus prognostically generated inundation.

To explore the temperature sensitivities of each GCP model, we fit a $Q_{10}$ curve for each GCP model (Fig. 5). The $Q_{10}$ parameter represents the sensitivity of wetland CH$_4$ fluxes to a 10°C increase in temperature, which provides insight into how strongly each model responds to temperature changes. A higher $Q_{10}$ value indicates that the flux estimates are more prone to change with temperature variations. Our analysis indicates a large variation in temperature sensitivity across the prognostic and diagnostic GCP models, but there is not a strong relationship between the magnitude of wetland CH$_4$ fluxes estimated by these models and the estimated $Q_{10}$ values (Fig. 5). We find the ELM has the lowest $Q_{10}$ value of all models at 1.77, suggesting that CH$_4$ fluxes in ELM are relatively insensitive to temperature changes compared to other models. In contrast, most of the other prognostic and diagnostic GCP models exhibit $Q_{10}$ values greater than 2, with the prognostic ISAM model showing the highest $Q_{10}$ of 11.92, suggesting a stronger temperature dependence. However, we do not find any correlation between wetland CH$_4$ fluxes from the GCP models and $Q_{10}$ values, meaning that models with the highest wetland CH$_4$ fluxes do not always have the





highest temperature sensitivity. As a result, $Q_{10}$ does not seem to be the most important contributor driving differences in the flux magnitude of the GCP models.

We also find that uncertainties in wetland area and inundation likely contribute to but are not the primary the cause of these
disagreements in flux magnitude. For example, the prognostic and diagnostic models usually yield a similar magnitude of fluxes, in spite of the fact that these different experiments do not use the same inundation estimates (Fig. 2). For Canada, the average total flux from the prognostic models is similar to the diagnostic models – 14.19 and 12.17 Tg per year, respectively (using GSWP3 meteorology). Similarly, the average total flux from the prognostic versus diagnostic models is nearly identical for the boreal forest biome. In some regions, the diagnostic models show greater agreement on the total annual flux than the
prognostic models, but in other regions, the prognostic and diagnostic models show similar levels of inter-model agreement (Fig. 2).

Interesting, we find models with simpler flux calculations yield flux magnitudes that agree more with atmospheric observations compared to those using more complex equations. GCP models such as LPJ-wsl, SDGVM, and JULES produce smaller flux magnitudes, and each of these models uses simple approaches to simulate $CH_4$ fluxes. For example, these models rely
only on net fluxes without accounting for specific transport pathways (e.g., ebullition, diffusion, or plant-mediated transport) (Zhang et al., 2024). In contrast, models such as VISIT, JSBACH, and ISAM have the largest flux magnitudes, and each of these models employs more complex equations that include multiple components of $CH_4$ fluxes, such as gross production, oxidation, and consumption. These models also simulate explicit transport pathways like ebullition, diffusion, and plant-mediated transport, alongside layered soil temperature schemes for temperature sensitivity (Zhang et al., 2024). Models with more com-
plex representations generally require additional input data to provide detailed flux estimates. This pattern suggests that the additional complexity in VISIT, JSBACH, and ISAM may introduce greater uncertainty in regions with more uncertain input data.

### 3.3   Seasonality

We find that models more consistent with atmospheric observations have a distinct seasonal peak in wetland $CH_4$ fluxes in July
and August. In contrast, models that do not agree well with atmospheric observations have a flatter seasonal cycle.

To evaluate these differences, we compare the correlation between atmospheric $CH_4$ observations and STILT simulations using each of the different GCP models (Fig. 6). We specifically use this analysis to explore which GCP models better capture seasonal and spatial variability of $CH_4$ fluxes across our model domain. First, we calculate $R^2$ values for each model using a two-predictor regression model. In each regression, the first predictor variables represents modeled $CH_4$ mixing ratios due
to wetlands using one of the GCP models, and the second predictor variable represents modeled $CH_4$ mixing ratios due to different anthropogenic flux products plus biomass burning from GFED (Sects. 2.3 and 2.4). The regression will scale the magnitude of the STILT model outputs to optimally match atmospheric observations. As a result, this analysis is not very sensitive to the absolute magnitude of the original flux estimates. Instead, the overall fit of each regression is more likely a reflection of the seasonal and spatial patterns in the wetland, anthropogenic, and biomass burning flux estimates; GCP flux
estimates with more accurate seasonal and spatial variability will more likely yield higher correlation coefficients ($R^2$ values).



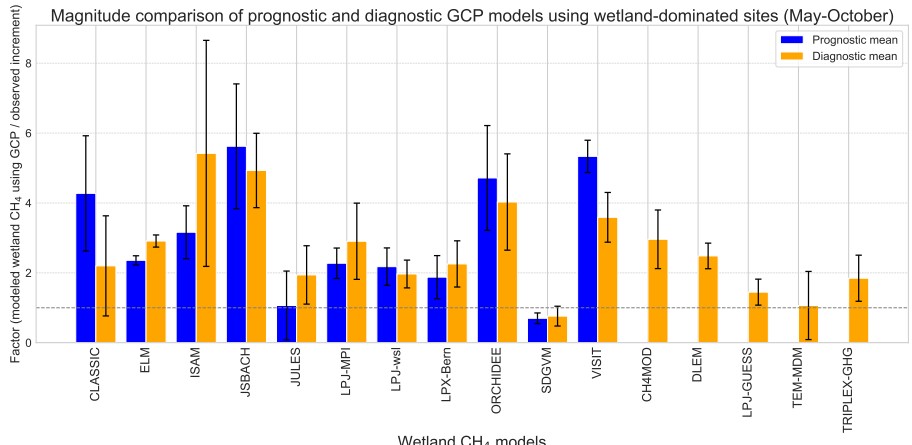

**Figure 4.** Comparisons between modeled mixing ratios from STILT against observations at the tower sites. The y-axis has values range from 0 to 9, representing the ratio between the modeled wetland $CH_4$ mixing ratios using the GCP models and the observed increment. We define the observed increment as the difference between atmospheric $CH_4$ observations and the sum of the boundary $CH_4$ levels, modeled anthropogenic $CH_4$ mixing ratios, and modeled biomass burning $CH_4$ mixing ratios. A value of 1 on the y-axis indicates perfect agreement between the modeled wetland $CH_4$ mixing ratios and the observed increment.

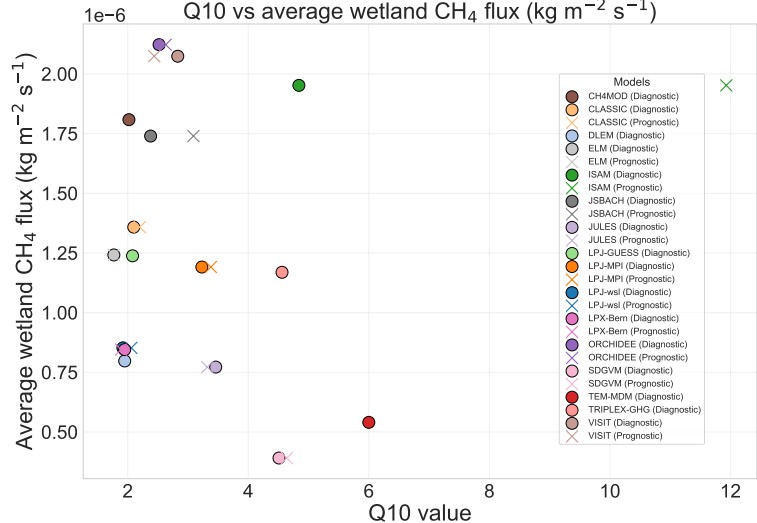

**Figure 5.** The plot shows the $Q_{10}$ factors estimated for each of the GCP models. Each colored shape represents an unique GCP model, and prognostic and diagnostic values are plotted separately for each model. The plot also shows the relationship between the magnitude of fluxes estimated by each model for the study domain and the $Q_{10}$ value estimated for each model.

Figure 6 depicts the mean $R^2$ values for 16 GCP diagnostic wetland models and 11 GCP prognostic wetland models. Each



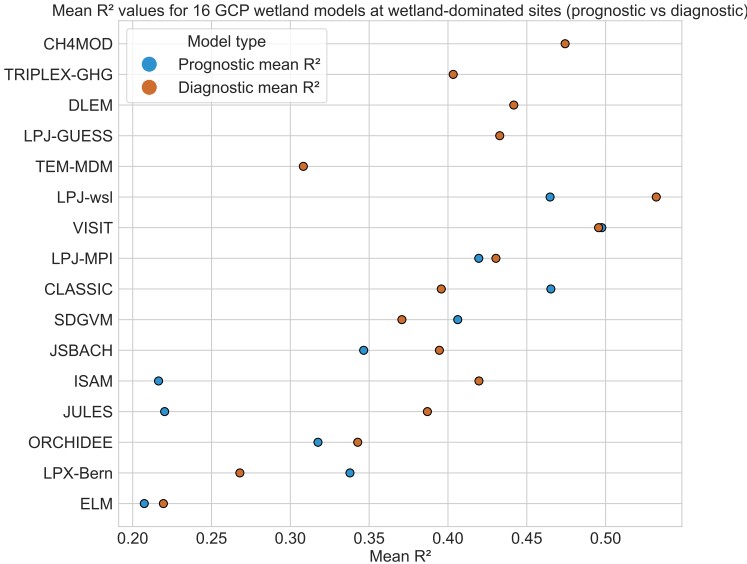

**Figure 6.** The correlation $R^2$ between modeled $CH_4$ mixing ratios using the GCP models and atmospheric observations. Blue dots represent the mean $R^2$ value for prognostic models across different climate forcing data and anthropogenic products. Orange dots represent the mean $R^2$ value for diagnostic models across different climate forcing data and anthropogenic products. The y-axis lists all the prognostic and diagnostic GCP models, and x-axis shows the $R^2$ range for these GCP models.

model has a mean $R^2$ value that is averaged from the two climate forcing data (GSWP3 and CRU) and three anthropogenic flux products. These results highlight the large variability in $R^2$ values across different GCP models.

Based on this analysis, we categorize each of the diagnostic and prognostic GCP models into three groups based on how they agree with atmospheric observations. By grouping the models, we can look for common patterns that separate models that exhibit high $R^2$ values from those that exhibit lower $R^2$ values. Models with $R^2$ values greater than 0.4 are grouped into the high $R^2$ group (represented by blue lines in Figs. 7a and 7b), models with $R^2$ values between 0.3 and 0.4 are classified as the average $R^2$ group (represented by green lines in Figs. 7a and 7b), and models with $R^2$ values below 0.3 are considered as the low $R^2$ group (represented by red lines in Figs. 7a and 7b). Although these cut-offs are inherently subjective, they offer a practical framework for grouping the models and result in a similar number of models within each group.

Across the high and average $R^2$ groups, $CH_4$ fluxes exhibit a clear seasonal cycle, and we find that approximately 60–70% of the total fluxes from these models during the period of May to October occur during the peak summer season (June, July, and August). In these groups, the models capture the sharp rise and fall of the $CH_4$ fluxes, and they also show peak monthly percentages during July and August (Figs. 7a and 7b). The low $R^2$ models display a much flatter seasonal pattern. The flatter seasonal cycle indicates that these models do not capture the pronounced summer peaks observed in the high and average groups, suggesting that they may not fully capture seasonal variations in wetland fluxes.



The relationships between $CH_4$ fluxes and temperature may explain some, though not all, of the differences in seasonality among the GCP models. In our study, diagnostic SDGVM, diagnostic LPJ-MPI, diagnostic JULES, and diagnostic ISAM are the models that have high and average $R^2$ values (>0.35), and both have estimated $Q_{10}$ values greater than three, indicating a high sensitivity of their fluxes to temperature changes. Moreover, models in the low $R^2$ group (<0.30) have estimated $Q_{10}$ values below 2, resulting in weaker temperature-driven flux variations (Fig. 5). This result shows that temperature relationships can explain at least some differences in the seasonality of the diagnostic GCP models. By comparison, existing empirical studies find a range of $Q_{10}$ values for wetlands in the arctic region. Cao et al. (1996) suggest that a $Q_{10}$ value of 2 is calculated using a simple temperature response model, but Ito (2019); Walter and Heimann (2000) compute the $Q_{10}$ values of 3.85 and 6 using a more complicated mechanistic temperature response model. In addition, another study finds that the composition of wetlands can also yield different $Q_{10}$ values in the arctic region. Specifically, M. Lupascu and Pancost (2012) find that wetlands that contain more Sphagnum moss can result in a $Q_{10}$ value of 8 or higher. These studies show that $Q_{10}$ values can be highly dynamic in high-latitude regions, and a $Q_{10}$ value of 6 does not necessarily mean that the temperature response model is wrong.

## 3.4 Spatial Distribution

We find that prognostic models that are most consistent with atmospheric observations concentrate their fluxes near the HBL (Fig. 8a). In contrast, prognostic models with the lowest $R^2$ values focus their fluxes outside this key region (Fig. 8c). To gain insight into the spatial patterns of prognostic GCP models, we analyze how their flux estimates vary across different regions. We focus this section on the prognostic models because the diagnostic models use the same inundation map and therefore exhibit similar spatial flux patterns. Similar to the previous analysis of seasonality, we group the prognostic models into three categories (high, average, low) depending on their $R^2$ values when compared against atmospheric observations. A Principal Component Analysis (PCA) highlights common spatial patterns among the models in each different group (e.g., Wold et al., 1987; Jolliffe, 1986; Delwiche et al., 2021). The percentage of variance explained by the first principal component shows the degree of spatial patterns shared among models in each group, and this percentage captures how consistently the models agree in their spatial flux distributions across grid boxes within the study domain. We find that models in the high $R^2$ group have a first principal component (PC1) explaining 63.5% of the variance, followed by the average $R^2$ group with 50.1%, and the low $R^2$ group with 68.9% explained variance. Although the low $R^2$ group shows the highest explained variance, this number does not necessarily indicate that the models in this group are more accurately capturing the true spatial patterns of the $CH_4$ fluxes compared to those in other groups.

We find notable common spatial features among the models in the high $R^2$, as seen in the PCA analysis. LPJ-wsl and CLASSIC have the highest $R^2$ values, and these models consistently concentrate their $CH_4$ fluxes in the HBL. In contrast, JULES, ISAM, and ELM are the models with lower $R^2$ values. These models show large spatial discrepancies in critical wetland regions such as the HBL, and they tend to concentrate fluxes outside of these key regions, particularly in the Great Lakes region of Canada.

An important caveat of this result is that the long-term observation network is sensitive to fluxes from some regions of high-latitude North America but not others (Fig. 1), so this analysis of spatial distribution is weighted to areas with good



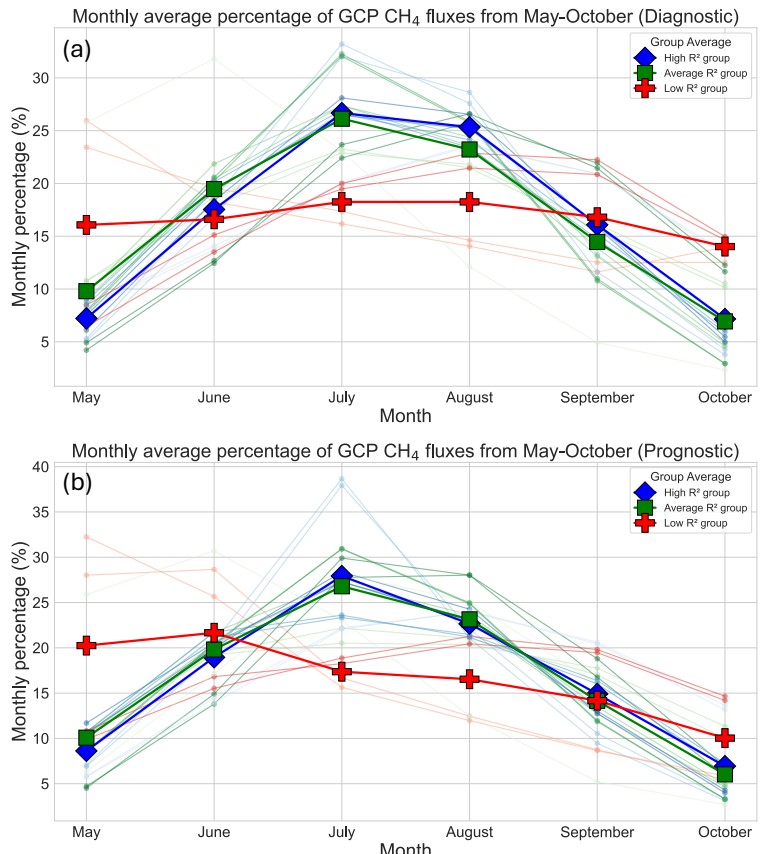

**Figure 7.** The seasonal cycles of the diagnostic GCP models (a) and prognostic GCP models (b) from 2007-2017. The blue, green, and red lines each represent the GCP models that have the highest, average, low $R^2$ values with atmospheric observations. The x-axis represents the months from May to October throughout 2007-2017, and y-axis denotes the percentages of $CH_4$ fluxes that occur within that month.

observational coverage. We also note that none of the atmospheric observing towers are directly located in the HBL, but the STILT footprints shown in Fig. 1 indicates that the network is sensitive to $CH_4$ fluxes from the broader region, allowing us to draw conclusions about the spatial distribution of fluxes in and around the HBL.

Interestingly, we also find that for 64% (7/11) of the models, the diagnostic version of the model yields a better fit ($R^2$) against
atmospheric observations compared to the prognostic version of the model (Fig. 6. Prognostic versions of CLASSIC, SDGVM, LPX-Bern, and VISIT have better $R^2$ values compared to diagnostic versions). The diagnostic and prognostic versions of each model often exhibit similar seasonal cycles (Fig. 7) but often exhibit different spatial patterns. This result suggests that the diagnostic inundation map is likely a more reliable or accurate inundation product than the inundation maps generated internally by the prognostic models, thus allowing the models to better capture regional $CH_4$ fluxes in high-latitude North
America.



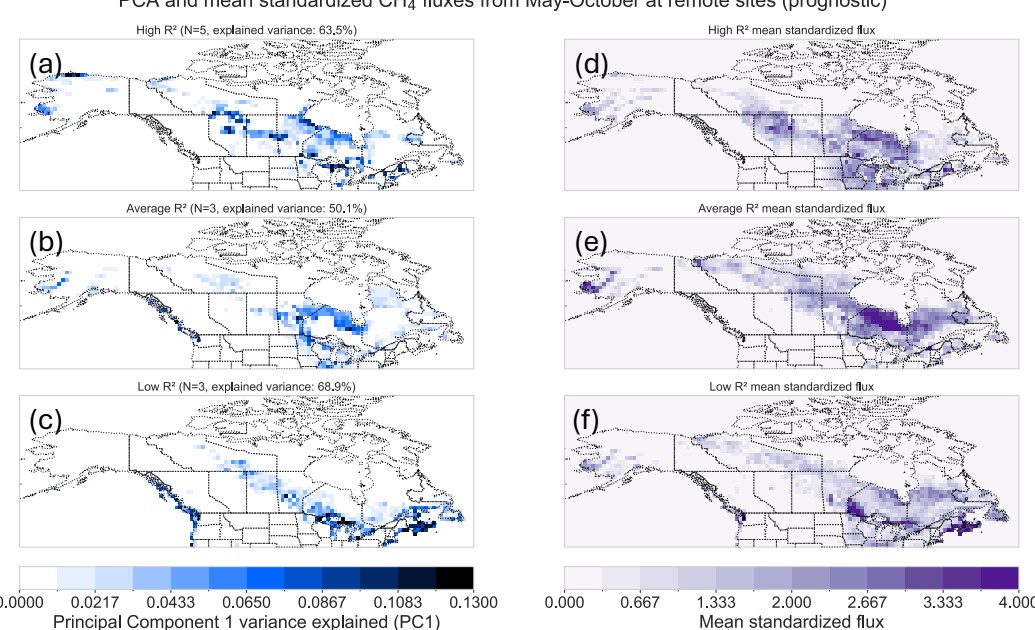

**Figure 8.** The PCA results and mean standardized CH$_4$ fluxes for the prognostic GCP models, run separately for each group of models – the high (a and d), average (b and e), and low (c and f) $R^2$ groups. The unit for PCA results is in explained variance by the first component (%), and darker (or more blue) shades represent better spatial agreements among the models within a same group.

## 4    Conclusions

This study highlights areas of convergence and disagreement among state-of-the-art process models of wetland CH$_4$ fluxes. We compare the estimates with atmospheric CH$_4$ observations between May and October in high-latitude North America. In the first section of the paper, we find that GCP models have a much smaller flux magnitude and lower inter-model uncertainty across North America compared to a previous model inter-comparison (WETCHIMP). This change in magnitude improves consistency with atmospheric CH$_4$ observations, though we argue that the current GCP model ensemble is still too high across much of Canada and Alaska. In the second section of the study, we find that process-based CH$_4$ models that are most consistent with atmospheric observations exhibit the highest percentage of fluxes in July and August relative to other months and have a sharper seasonal cycle. These process-based models also concentrate their fluxes near the HBL while less skilled models often concentrate fluxes further south near the Great Lakes.

Overall, this study highlights the opportunity to improve current process-based models to estimate regional wetland CH$_4$ fluxes. Key areas for improvement in model parameterization include addressing uncertainties in inundation maps to capture wetland extent and improving estimated maps of soil carbon, though the latter factor was difficult to evaluate this study. We find that prognostic models show greater room for improvement than the diagnostic models; while diagnostic models benefit from consistent inundation maps, the development of better prognostic models is nevertheless very important because these



models can be used to project future trends in wetland extent or inundation, which is critical for future projections of $CH_4$ fluxes under the ongoing climate change. Overall, we argue that the bottom-up modeling community had made large strides in reducing inter-model uncertainties, and these improvements are consistent with atmospheric $CH_4$ observations. With that said, there is still an enormous need for further improvements in these models to advance understanding of high-latitude wetland
$CH_4$ fluxes in a changing climate.





*Data availability.* We received the wetland model estimates from Zhen Zhang and the GCP modeling team, and these datasets are available upon request from the GCP modeling team. The GlobalViewPlus $CH_4$ ObsPack v5.1 dataset is available at https://gml.noaa.gov/ccgg/obspack/citation.php?product=obspack_ch4_1_GLOBALVIEWplus_v5.1_2023-03-08.

The WRF-STILT footprints for North American $CH_4$ monitoring sites are available at https://gml.noaa.gov/aftp/products/carbontracker/
lagrange/footprints/ctl-na-v1.1/. The North American Boundary Condition product is provided by the NOAA Earth System Research Laboratory, and the dataset is available at https://gml.noaa.gov/aftp/user/arlyn/naboundary/v20190806/ROBJ/. Guidance related to these datasets can be requested from Arlyn Andrews (Arlyn.Andrews@noaa.gov) and Kathryn McKain (Kathryn.McKain@noaa.gov).

The CAMS global emission inventory dataset is available from the Copernicus Atmosphere Data Store. DOI:https://doi.org/10.24381/1d158bec. CarbonTracker CT-$CH_4$-2023 data are available from NOAA's Global Monitoring Laboratory. DOI:https://doi.org/10.25925/
40jt-qd67. The gridded inventory of Canada's anthropogenic $CH_4$ fluxes is available from the Harvard Dataverse. https://doi.org/10.7910/DVN/CC3KLO. The gridded U.S. Greenhous Gas Inventory (Version 2) can be found on Zenodo. DOI: https://doi.org/10.5281/zenodo.8367082. The Global Fire Emissions Database, Version 4 (GFEDv4) is available through the Oak Ridge National Laboratory (ORNL) Distributed Active Archive Center (DAAC). DOI: https://doi.org/10.3334/ORNLDAAC/1293.

*Author contributions.* HL and SMM designed the study and wrote the manuscript. FRV, MI, ZZ, BP, JRM, LF, ALGL, AC, ZH, DCG, DC,
VY, and DH provided feedback and comments on the manuscript. LF, AC, ZH, and DCG provided modeling support. JRM provided the WETCHIMP models. ZZ, BP, and the GCP modeling team provided the prognostic and diagnostic process-based $CH_4$ flux models. FRV, MI, DEJW, and DC contributed to the collection and maintenance of Canadian in situ tower $CH_4$ measurements included in the NOAA ObsPack data product. ALGL and DH provided valuable suggestions on the $Q_{10}$ section.

*Competing interests.* The authors declare that they have no conflict of interest.

*Acknowledgements.* This work is funded by NASA ABoVE grant (#80NSSC22K1246) and by an NSF CAREER award (#2237404). We thank Environment And Climate Change Canada (ECCC) and NOAA Global Monitoring Laboratory for providing the GLOBALVIEWplus $CH_4$ ObsPack v5.1 dataset that is important for the completion of this work. We also acknowledge the use of WRF-STILT footprints data that are produced as a part of the CarbonTracker-Lagrange project with the support from NOAA's Climate Program Office and NASA's Carbon Monitoring System. We also acknowledge the use of the NOAA Earth System Research Laboratory's North American Boundary Condition
product for $CH_4$, and we thank Arlyn Andrews, Kathryn McKain, and all other collaborators for providing access to the dataset. We use the anthropogenic $CH_4$ emissions from the 2020 CAMS global emission inventory. This work contains modified Copernicus Atmosphere Monitoring Service information [2020]. Neither the European Commission nor ECMWF is responsible for any use that may be made of the Copernicus information or data it contains. The CarbonTracker CT-$CH_4$-2023 results are provided by NOAA GML, Boulder, Colorado, USA from the website at https://gml.noaa.gov/ccgg/carbontracker-ch4/. We acknowledge the use of the fire emissions from the Global Fire
Emissions Database version 4 (GFED4s) described in van der Werf et al. (2017), and we regrid the dataset for this work. We also thank the Global Carbon Project (GCP) modeling team for their invaluable contributions in developing the models. In addition, we specifically





thank the following individuals for their important work in building the GCP models: Benjamin Poulter, Philippe Ciais, Joe Melton, William Riley, David Beerling, Nicola Gedney, Peter Hopcroft, Akihiko Ito, Atul Jain, Fortunat Joos, Thomas Kleinen, Tingting Li, Xiangyu Liu, Paul Miller, Changhui Peng, Shushi Peng, Zhangcai Qin, Qing Sun, Hanqin Tian, Yi Xi, Wenxin Zhang, Qing Zhu, Qiuan Zhu, and Qianlai
Zhuang.



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
