# Peer review of "Methane fluxes from Arctic & boreal North America: Comparisons between process-based estimates and atmospheric observations"

_EGUsphere, 2025_

## Author Comment (AC1)

We thank the reviewers for their thoughtful and constructive suggestions. These comments were very helpful during the revision. We have incorporated the requested changes and clarifications throughout the manuscript. Below, each reviewer's comment appears in bold and our reply follows in blue italics.

**Response to reviewer #1:**

**#1:**

This study presents an update of the WetChimp wetland model intercomparison that was published several years ago. The new inter-comparison makes use of model submissions to the Global Carbon Project. The results show a significant reduction in inter-model spread compared with the previous intercomparison, in closer agreement with atmospheric measurements evaluated using Stilt over North America. This is regarded as a sign of good progress in developing these models. In my view, explained below, this should consider another possible explanation. The comparison with WetChimp is only indirect, since the results were not included in the evaluation using atmospheric measurements. Furthermore, the evaluation using atmospheric data concentrates on R2, for a reason that remains unclear. After these concerns are repaired and accounted for, I see no reason to uphold publication of a study that could provide a useful new reference.

Response: Thank you for the feedback. We have now included WETCHIMP models in the analysis. Specifically, we incorporate the WETCHIMP comparisons into the R2 plot, showing how WETCHIMP models align to our atmospheric observations vs how GCP models align with our observations (Figure 5). We have also modified the main text by adding a WETCHIMP section under Methods, and we have also added a couple sentences to describe these differences (in Sect. 2.2). Regarding the R2 value, we also added a figure in the supplement (Figure S10 and S11) to show the RMSE of these models, and we have shown that the ranking of the models using RMSE is similar to the R2 metric.

**GENERAL COMMENTS**

The risk of model intercomparisons is that they might steer model development in the direction of the "mean model". It is tempting to interpret a convergence in model results as progress towards uncertainty reduction. This is only true, however, if the models converge to the true state. The evaluation that is presented does not provide evidence that this is the case.

Response: We agree that reduced inter-model spread does not by itself demonstrate improved accuracy. In the revision, we fix the sentences in the last paragraph of section 3.1 [(Lines 293–296)] acknowledging that reduced inter-model spread does not necessarily mean improved accuracy because convergence can happen due to shared inputs, similar design choices, or the number of the ensemble models, and we also emphasize that

accuracy is assessed with the atmospheric evaluation rather than spread alone (Figure 5). When we use the same ensemble (Bern, WSL, SDGVM, DLEM, Orchidee), WETCHIMP models still show a higher averaged annual flux total (20.46 Tg CH4/year vs 15.14 Tg CH4/year) compared to the GCP models (Figure S5 and S6). As a result, the convergence yields smaller values across these flux models, which is in better agreement with atmospheric CH4 observations.

Atmospheric measurements are used to test the quality of wetland emission estimates. But, for a reason that is not clear, they are not used to confirm that the WetChimp submissions are less realistic. The argument that they are is only based on the convergence of results and the size of the emissions. The analysis of new submissions suggests that models with lower emissions are more accurate, based on the amplitude of concentration increments, but the argument is again rather indirect as this comparison also did not include the WetChimp emission estimates. I propose to either redo the analysis using the WetChimp fluxes, or – if that is not possible – acknowledge this short coming of the method that is used.

Response: Thank you for the suggestion. We agree with this argument and have added the WETCHIMP comparisons with a figure in the supplement (Figure S10 and 11) and sentences mentioned in [(Lines 418–423)]. Note that WETCHIMP model simulations only extend through the year 2004, which does not overlap the same study time periods from 2007 to 2017. However, we assume that CH4 fluxes do not vary much from 2004 to 2007, so we use 2004 as a sample year and run the WETCHIMP models through STILT for comparisons. When using the same subset of models between GCP (prognostic and diagnostic) and WETCHIMP (LPX-Bern, ORCHIDEE, LPJ-wsl, and SDGVM), the mean R² for the WETCHIMP models are lower than the GCP models, and the mean RMSE for the WETCHIMP models is higher than the GCP models (Figure S10). When including all models between GCP (prognostic and diagnostic) and WETCHIMP ensembles, the mean R² for the WETCHIMP models are still lower than the GCP models, and the mean RMSE for the WETCHIMP models is still higher than the GCP models (Figure S11). Overall, our results indicate that the GCP model ensembles have better agreement with the atmospheric observations compared to the WETCHIMP model ensemble.

The model evaluation method uses R2 as a metric of agreement with the observations. R2 is limited, however, in that it does not penalize a wrong enhancement amplitude. The observed concentration variability is explained mostly by the weather. Differences in emissions show up rather in the concentration increments, which are not captured by R2. A more logical choice would have been to use RMSE as evaluation metric. This should either be tried, or an explanation should be given of why it was not done. Note that RMSE is not the same as the metric shown in Figure 4, although that does provide an evaluation based on the size of the mean concentration increment.

Response: Thank you for the feedback. We now add the RMSE plot in the supplement (Figure S7) and clarify that we use R2 metric rather than RMSE because the model rankings are the same when using these approaches (second paragraph in section 3.3, [(Lines 378–380)]. However, we include the RMSE plot and analysis anyways to show that we have done this to explain why we stick to the R2 metric.

Based on the results in Figure 6, it is suggested that simpler diagnostic models perform better than more sophisticated prognostic models. This raises the question, however, how independent the model results are of the data that are used to evaluate them. Simpler models are easier tuned to the existing measurements than sophisticated mechanistic models. Could that explain why they score better? I was surprised to see that the evaluation is based only on ambient air measurements, without the mentioning of flux measurements that are made at several sites in the study domain. They might even provide a less independent means of evaluation. It would nevertheless provide useful additional information to compare the performance of the different model categories that are distinguished.

Response: We added a clear caveat in the sixth paragraph of section 3.3 [(Lines 409–417)], stating that the higher apparent skill of diagnostic runs may partly reflect their reliance on the consistent inundation product. In addition, simple models do a good job at capturing regional-to-continental scale flux patterns as effectively as more complex models (e.g., Miller et al., 2014, 2016b), whereas additional process complexity may become increasingly important for simulating finer-scale spatiotemporal variability. We do not incorporate eddy flux observations because they are sensitive to very different scales of spatial-temporal variability than atmospheric observations. Atmospheric observations are usually sensitive to variability in fluxes that occur over 100s of kilometers in spatial scale, whereas eddy flux observations are sensitive to variability of fluxes that occur over a few kilometers in spatial scale. Our goal in this study is to evaluate these process-based models across broad regional to continental spatial scales. In addition, a comparison against eddy flux observations could be challenging because many of these models have a spatial resolution of 10,000 km² (i.e., ~100 km x 100 km), whereas many eddy flux observations have footprints of approximately 1 km².

**SPECIFIC COMMENTS**

Line 75, how about regional models for the study domain? I understand that this model inter-comparison evaluates global models, but results from regional models might nevertheless provide useful information for evaluating them.

Response: Thank you for the feedback. We are not aware of regional model inter-comparisons or model ensembles that focus on high-latitude North America. The GCP does not include any regional simulations in their process-based model ensemble, and no regional simulations are available from WETCHIMP either. Coordinating a regional, process-based model inter-comparison for high-latitude North America is really beyond what we can do in a single paper focused on atmospheric observations.

Line 94, the purpose of this sentence in relation to the previous is not clear. Is it meant to provide further justification for afternoon measurements? Or is it meant to indicate a limitation that will anyway play a role? Please rephrase to clarify.

Response: Thank you for pointing this out. We use the atmospheric observations during afternoon hours because data in this time period have less transport errors tied to uncertain planetary boundary layer (PBL) height. By contrast, atmospheric models like WRF-STILT sometimes have difficulty accurately modeling stable nocturnal PBL dynamics. This practice of using afternoon-only observations is common in regional/top-down evaluations, and many studies have used this approach (We mention these studies in the third paragraph of section 2.1, [(Lines 107–113)]).

Line 99, Don't the campaigns in Alaska offer a useful opportunity for further validation? If so, why was it not used?

Response: We agree that the Alaska aircraft campaigns are valuable. Several existing studies compare process-based  $CH_4$  flux models against aircraft observation collected in Alaska, including Chang et al. 2014, Miller et al. 2016, and Hartery et al. 2018. We have added text to the discussion section to compare and contrast our results with the results of those studies (lines 317–321 & lines 391–392). Existing, intensive aircraft campaigns are geographically concentrated in specific regions of Alaska (e.g., the Yukon Kuskokwim Delta, the Utqiagvik region) and in specific years, whereas our analysis spans high-latitude North America for about a decade. We are currently working on a separate study focused on geostatistical inverse modeling of  $CH_4$  fluxes in the aforementioned regions of Alaska using intensive aircraft observations.

Line 168: From a simple back of the envelope calculation it seems the 1 - 1.5 ppb represents high-latitudes already, because the global decay due to OH should be faster.

Response: 1 to 1.5 ppb does reflect global  $CH_4$  loss due to OH over 2-3 days timescale (lifetime-based estimate, not a regional OH sink), coincident with the period of strongest STILT sensitivity (Miller et al. (2013)). Using the globally assessed  $CH_4$  lifetime (9-12 years; e.g., Prather, 2012; IPCC AR6), the expected first-order decay over the 2-3 day period of strongest footprint sensitivity is less or equal to 1-2 ppb for a ~1900 ppb background, which is consistent with the < 1-1.5 ppb back-of-envelope estimate used for context. In addition, we fit our boundary conditions to regional free-troposphere values, which accounts for

long-range oxidation processes already. We have included this clarification in the third paragraph of section 2.4 [(Lines 183–190)].

Figure 1: Does 'daily' mean that the footprints shown in this figure represent only the influence of a one day back trajectory? The text mentions that 10-day back trajectories are used, which raises the question why mean 1-day footprints are shown here. Is the 'mean' evaluated over 2007 – 2017 (if so then this should be mentioned explicitly).

Response: We have revised the caption and the text to explicitly state that Figure 1 represents the mean 10-day (0-240h) integrated footprints averaged over 2007 to 2017. We have removed the "daily mean footprints" to avoid confusion.

Line 191: "The remaining sites ..." You might want to add a reference to Figure 1 where these sites are indicated as red circles.

Response: We have added the reference to Figure 1 when listing the excluded and included tower sites (Last paragraph of section 2.4, [(Lines 218-225)]).

Line 205: Did you test how reliably the apparent Q10 approximates Q10 for the models that use a Q10 formulation? (and for which its value is known)

Response: The apparent Q10 plot is shown in Fig. S9 in Zhen Zhang's 2025 paper. Zhang et al. report that the mean Q10 value for the global model ensemble (June–August) is 2.1. In comparison, our mean Q10 value for the model ensemble over high-latitude North America is 3.21, which is reasonable because colder ecosystems often exhibit stronger apparent temperature sensitivity. In addition, we reviewed previous studies that reported Q10 values for the GCP models, and we find that different authors sometimes report different Q10 values for the same model. For example, Hopcroft et al. (2014) report a Q10 value of 1.5 for SDGVM (Table 2), whereas Ringeval et al. (2013) describe a value of 3 for the same model. Moreover, none of the existing studies report Q10 values specifically for our study domain, which makes direct comparison between their apparent Q10 values and ours difficult.

Line 215: How about the seasonality of anthropogenic emissions?

Response: To clarify, the anthropogenic CH4 emissions in our analysis are not from our own estimates but are derived from three established anthropogenic flux products, each of which is based on inventories or data assimilation systems. Specifically, we use: (1) CarbonTracker CH4 2023 (Oh et al., 2023); 2) a combined gridded inventory of the U.S. Greenhouse Gas Inventory (Version 2) and Canada's anthropogenic CH4 fluxes (Monforti Ferrario et al., 2021; Maasakkers et al., 2023; Scarpelli et al., 2021); and (3) the Copernicus Atmosphere Monitoring Service (CAMS; Granier et al., 2019). We have added a plot in the supplement (Figure S8) to show the seasonality of the modeled CH4 mixing ratios using each of these anthropogenic products. Each color represents an unique anthropogenic

product, and the figure shows the monthly contribution of anthropogenic CH₄ mixing ratios (in ppb) to the total simulated mixing ratios (derived from STILT run) from May to October.

Line 220: But anthropogenic emissions inventories provide estimates for each year, so reasonably accurate IAV estimates exist for the anthropogenic part.

Response: We agree that year-specific anthropogenic inventories exist. However, previous studies argue that these inventories are uncertain and likely underestimate emissions by 50-100%, especially in oil and gas producing regions of western Canada (e.g., Chan et al. 2020, Ishizawa et al. 2024, MacKay et al. 2021). We try to mitigate the impacts of the uncertainties in anthropogenic emissions on our analysis through two means: (1) We analyze sites that are distant from major anthropogenic sources and are instead dominated by wetland emissions; (2) We use multiple anthropogenic emissions estimates from both bottom-up and top-down studies to better characterize the uncertainties in our analysis due to uncertain anthropogenic emissions. In spite of these measures, we worry that uncertainties in IAV from anthropogenic sources, particularly the oil and gas industry, could make it challenging to isolate IAV in wetland sources. Canada's oil production changed by ~53% between 2007 and 2017 (data source: https://ourworldindata.org/fossil-fuels), and the resulting changes in methane emissions from this source could be difficult to disentangle from year-to-year variability in wetland methane emissions.

Line 227: Could it be that WetChimp led to a consensus about the mean flux that might explain some degree of convergence?

Response: Community intercomparisons like WETCHIMP can encourage convergence toward a common magnitude independent of the truth. In our analysis, we address this point using our results based on the atmospheric evaluation (R2 metric in figure 5) rather than reduced inter-model spread alone.

Line 229: Is this also true for the models that are common to both experiments?

Response: We have added a plot in the supplement (Figure S6) showing the annual  $CH_4$  flux total by the GCP models. There are five overlapping models from both experiments (Bern, DLEM, Orchidee, WSL, SDGVM), and the mean annual flux total by the WETCHIMP models are still ~4 Tg  $CH_4$  per year higher than the mean annual flux total by the GCP models. As a result, the GCP models exhibit smaller flux totals not because they are using more models than the WETCHIMP ensemble.

Line 275: How are emissions from fresh water accounted for in the current study?

Response: We use the HEMCO software to process CH4 flux inputs. Freshwater emissions from lakes and reservoirs are included through the LAKES inventory in HEMCO, which

reads the 'Lakes' dataset (Maasakkers et al. 2016) provided with the GEOS-Chem emissions package (available at: GEOS-Chem data archive). We have added a paragraph in section 2.4 to make further clarification on this question ([(Lines 201–207, 300-302)])).

Figure 2: An explanation about the error bar should be added in the figure caption.

Response: We have revised the caption in figure 2 and defined the uncertainty bars.

Line 315-317: It is not clear why Q10 would correlate with the average methane emission (which indeed seems not to be the case). Wouldn't it have been more logical to assess Q10 against R2 or against the seasonal amplitude?

Response: Following your comment, we compared each model's Q10 value against its mean R² when evaluated against atmospheric observations (Figure S13). We do not find a strong or consistent relationship between Q10 and R². Although models with the highest R² values have relatively lower Q10 values of <3 (LPJ-wsl, VISIT), models with the lowest R² values also exhibit lower Q10 values of <2 (ELM). This result is consistent with the fact that Q10 reflects only the relative temperature sensitivity of fluxes, while model—data agreement also depends strongly on hydrology, substrate dynamics, and other processes. We therefore conclude that Q10 by itself is not the most important predictor of model performance, though it remains useful as a diagnostic for temperature response.

Line 357: It would be useful to add standard deviations to the points in figure 6 corresponding to the averages over climate forcing data and anthropogenic emission inventories.

Response: We added the standard deviations to the points in Figure 6 in our original plot, but we found it difficult to read and messy. To balance clarity and completeness, we decided to keep the current plot for a cleaner panel and use the averaged R2 values only.

Line 376: Figure 5 is referred to for a relation between Q10 and flux variations, but this figure relates Q10 to the mean flux rather than its variation.

Response: We have corrected the text to describe Figure S12 (Q10 vs mean flux).

Line 395-398: This rightly mentions that the explained variance of the PC1 has no relation with the true variance. However, more useful would have been to explain what the comparison of these numbers does mean. Right now, it is unclear why these numbers are even mentioned.

Response: We have now clarified this point in the revised text. The PC1 explained variance is used as a measure of within-group spatial coherence, or the fraction of between-model variance captured by the dominant common spatial pattern. A larger PC1 indicates that models in that group have more shared spatial patterns, but a larger PC1 does not

necessarily mean that their spatial patterns are closer to the truth (first paragraph of section 3.4, [(Lines 431–438)]).

Line 405: 'so this analysis of spatial distribution' It is not clear what 'this analysis' refers to. The PCR analysis is not weighted to areas with stronger observational coverage, is it?

Response: "This analysis" refers to the PCA analysis of the model fields, which is unweighted across grid cells. By contrast, our evaluation metric (spatial R²) is computed at the tower/aircraft locations, so our confidence in spatial skill is correspondingly higher in those observed regions.

Line 420: 'this change in magnitude improves ...' This cannot be concluded because the WetChimp flux estimates were not included in the comparison to observations.

Response: We have added a paragraph/section of the WETCHIMP comparisons (lines 454-456), and we find that the GCP model estimates are indeed more consistent with atmospheric observations compared to the WETCHIMP models (Figure S9, S10, and S11).

Line 422: 'most consistent with atmospheric observations' only concerns the R2, whereas it is not clear that R2 is best metric to evaluate the consistency with atmospheric observations.

Response: We have added both  $R^2$  and RMSE plots in our analysis (Figure S7), and both metrics show the same model rankings. As a result, we could possibly conclude that models that have highest  $R^2$  values are most consistent with atmospheric observations.

Line 432: 'Overall, we argue ...' It should be made clear that this conclusion only holds for the current analysis of emissions from Northern America. Since the models are global, there is still the possibility that other regions turn the overall outcome in the opposite direction.

Response: We have revised the last paragraph in the conclusion section, stating that our conclusions apply to the North America domain and period only, and the results may not generalize to other regions [(Lines 469–470)].

**TECHNICAL CORRECTIONS**

Line 180: 'initially' instead of 'preliminary'?

Response: We have now adopted 'initially' instead of 'preliminary'.

Line 324: "contribute to<o>" (?) but are not "the primary >the< cause"

Response: We have fixed the phrasing as: "We also find that uncertainties in wetland area and inundation likely contribute to but are not the primary cause of these disagreements in flux magnitude." [(Lines 437)]

**Response to reviewer #2:**

**2:**

General comments:

This paper is an interesting spinoff from the community-wide GCP Global methane budget effort, this time focused on the skill of the flux models for the arctic and boreal North America. It first compares a recent batch of simulations with an older one, then compares the former with atmospheric observations using a transport model. While the study is commendable, it often lacks subtlety, as detailed below, and, like many studies of this type, it does not really allow novel insights. It should certainly be published, but after a major revision.

Response: We appreciate your thoughtful feedback. We have revised the manuscript in response to each of the following comments.

**Detailed comments:**

I. 5, 419, 433: what is the "inter-model uncertainty"? I suspect a loose concept behind it.

Response: We have now defined "inter-model uncertainty" in the third paragraph of section 3.1 as the across-model standard deviation of the May to October mean wetland flux at each grid cell [(Lines 256–259)].

I. 12: HBL for "Hudson Bay Lowlands" is defined three times in the text, but actually I would encourage the authors not to abbreviate this region.

Response: We have now spelled out "Hudson Bay Lowlands" on the first use and avoid repeating the abbreviation.

I. 41-43: how does the use of "similar modeling protocols etc." allow us to identify and diagnose uncertainties in models? By construction the uniformization of some of the input data and configuration focuses the analysis on a subset of the uncertainty sources.

Response: By harmonizing input data ("similar modeling protocols"), the GCP modeling protocol minimizes input-driven variability, so that the remaining inter-model spread mainly reflects model-internal differences (such as process representation). This design lets us diagnose structural uncertainties across the models. In other words, these process-based models usually require meteorology or climate forcing data as an input from an external source (CRU or GSWP3). If all the models use the same climate forcing data (as has been done here), we can focus on differences among the flux models themselves without

confounding differences due to different meteorology and/or climate forcing input data. We have clarified this scope in the manuscript in the third paragraph of introduction [(Lines 42–46)].

I. 49: weird phrasing. Please reformulate.

Response: We have rephrased the sentence for clarity [(Lines 47–52)].

I. 52-61: "Although ... Notwithstanding" Back-and-forth reasoning. Please reformulate more linearly.

Response: We have restructured this paragraph accordingly (line 52-57).

I. 61: What does "narrower range of uncertainties" mean?

Response: By "narrower range of uncertainties," we mean that top-down inversion studies typically produce smaller posterior uncertainty ranges on regional CH4 flux estimates (i.e., the uncertainty bounds around the optimized fluxes after assimilating atmospheric data) [(Lines 61–64)].

I. 68-70: Now you need to say more: what did we learn from these studies?

Response: We have revised the sixth paragraph of the introduction to provide a clearer synthesis of what has been learned from these studies (for more details: [(Lines 65–76)]).

I. 74:77: trivial statement. Please remove.

Response: We have removed these lines.

I. 101: odd argument for leaving the aircraft data out here. What did we learn in these studies which is interesting for the present one? If nothing, then the job would still need to be done.

Response: Our intent was not to dismiss aircraft data. Previous aircraft-based inversions (e.g., CARVE and Arctic-CAP) have already done an in-depth analysis of process-based models for these specific subregions of Alaska (Y-K Delta and Utqiagvik) that have plentiful aircraft observations. These studies estimated growing-season (May–October) emissions of 1.7–2.3 Tg CH4 per year, with fluxes peaking in July–August rather than early summer months. They also revealed pronounced spatial heterogeneity: tundra ecosystems contribute a disproportionate share of Alaskan fluxes (more than 50% of the total CH4 fluxes in Alaska come from tundra ecosystems, and about 24% of the statewide fluxes come from

the North Slope), which is much higher than the estimates from process-based models. Long-term records from Utqiagvik also demonstrate that CH4 fluxes can persist into November–December, underscoring the importance of late-season fluxes. In contrast, our goal is to provide a broader evaluation of the GCP models across the entire high-latitude North America domain, leveraging the continuous tower network that offers wider spatial and temporal coverage. We have added discussions of the aircraft data to Sects. 1, 2.1, and 3 (i.e., the Results and discussion) [(Lines 58–71, 94-98, 317-321, 391-392)]).

Section 2: we are missing a subsection on WETCHIMP.

Response: We have added a discussion describing the WETCHIMP models in section 2.2 (Wetland CH4 flux model ensembles: GCP and WETCHIMP) [(Lines 132–139)].

I. 108: could you say more about how inundation is estimated in prognostic models? Is it really prognostic or observation-driven like in the diagnostic models?

Response: In the diagnostic simulations, an inundated area is prescribed from WAD2M v2. In the prognostic simulations, models estimate inundation internally using their own hydrologic schemes (e.g., through simulated water table depth, soil moisture, or surface runoff; see section 2.1 in Zhang et al., 2025). Thus, prognostic inundation is not observation-driven but generates from each model's land surface and hydrology formulation, and differences across models largely reflect these scheme-specific implementations. We have clarified this distinction in the manuscript in the first paragraph of section 2.2 [(Lines 117–122)].

I. 109: what is the point of saying that the GCP modeling groups submitted flux estimates to the GCP?

Response: We have now fixed this paragraph in section 2.2 and have removed the redundant text [(Lines 122–125)].

I. Section 2.3 we are missing some information about the temporal resolution and time range of this data

Response: We have added the temporal resolution and time range in section 2.3 of the revised manuscript [(Line 151)].

I. 145: how do you simulate fluxes with WRF-STILT?

Response: WRF–STILT does not simulate fluxes, but it simulates  $CH_4$  mixing ratios in the atmosphere (e.g., at an atmospheric observation site) using a flux estimate as an input into the model. In other words, WRF–STILT provides the transport information, and the fluxes come from the process-based models such as the GCP. We have now revised the first paragraph of section 2.4 for clarification [(Lines 165–173)].

I. 155: I understood that the study time frame stopped in December 2017.

Response: We have fixed this and clarified that the evaluation window ends in 2017 [(Line 174)].

I. 184-185: fine, but then don't write that you used a 1.5 threshold a few lines before.

Response: We have revised the language to state that we focus on tower sites where the ratio of modeled GCP to modeled anthropogenic  $CH_4$  mixing ratios is greater than 1.3 (not 1.5). We selected this threshold because the next-highest site, Abbotsford (ABT), is an urban location with a ratio of 1.06. Including sites with ratios near 1 would introduce urban locations that are more strongly influenced by anthropogenic emissions and could bias our results. In contrast, if we set the threshold to 1.5, then we would exclude some sites, like East Trout Lake (ETL) which is located in a sparsely populated wetland region. We have added additional explanations in lines 211–218.

I. 218-222: I understand that the discussion cannot be extensive as you say, but where is it? Please substantiate the statement based on your data, or simply say that you did not study IAV without speculating.

Response: We agree and we have removed speculative language and state that IAV is out of scope for this paper [(Lines 227–229)].

I. 226, 228: the use of "consensus" is ambiguous here because it only refers to converging numbers for whatever cause, not to "scientific consensus" among the modelers. Please use a non-ambiguous term.

Response: We have replaced the word "consensus" with "reduced inter-model spread" and state that reduced spread does not imply improved accuracy [(Lines 232–234, 293–296)].

Section 3.1. The statistical analysis is too short. You are comparing standard deviations estimated on ensembles made of few members only and of varying sizes. Further, some members are most likely correlated: in Table S2, there are three flavors of LPJ and I guess that most models share some parameterizations together. Given the importance of this section for the paper conclusions, you need to make it much more robust.

Response: To address this, we (i) added Figs. S5–S6 showing annual CH4 flux totals for the overlapping model set (LPX-Bern, DLEM, ORCHIDEE, LPJ-wsl, SDGVM). Using this matched subset, the WETCHIMP mean remains ~4 Tg of CH4 per year higher than the GCP mean in Canada [(Lines 284–289)]. (ii) We recomputed the inter-model uncertainty using the same shared subset, but WETCHIMP still exhibits a larger spread than GCP in Canada (Fig. S9) [(Lines 289–292)]. These checks confirm that our findings (lower flux magnitude and reduced inter-model spread in GCP) are consistent when using the same model ensemble.

I. 231: the definition of the error bars should also appear in the legend of Figure 2.

Response: We have now added the definition of the error bars in the legend of Fig. 2.

I. 263: all models could be wrong the same way and there would be opportunity for improvement. Please rephrase.

Response: We have now rephrased the sentence to show that even with agreement that there is still room for improvement [(Lines 268–270)].

I. 278-279: this reasoning ("This improved inter-model agreement implies... more accurate") is shocking.

Response: We have now revised this sentence to show that improved inter-model agreement does not necessarily indicate accuracy, and reduced spread only indicates greater consistency among models (e.g., via harmonized parameterizations). Our point is that the recent GCP ensemble shows that models have become more consistent with each other compared to the WETCHIMP ensemble, but this reduced inter-model spread is not a proof of model accuracy [(Lines 293–294)].

Figure 2: whiskers should be defined.

Response: We have now added the definition of the error bars in the caption of Figure 2.

I. 297: this sentence actually stems from the statement of I. 156 about the methane lifetime. Nothing new.

Response: We want to keep this sentence because we want to clarify that anthropogenic fluxes dominate in some of our study regions and can have an impact on separating them from wetland sources.

I. 324: one "the" too much

Response: We have corrected the wording in the revised manuscript [(Lines 346–347)].

I. 332: "Interestingly"

Response: We have fixed this based on your suggestion [(Line 354)].

I. 341: I am not following the logic here.

Response: We have rephrased this statement for clarity [(Lines 361–363)].

I. 413: didn't we know that beforehand?

Response: We have fixed this sentence now by showing the diagnostic models, when run through STILT, better agree with atmospheric observations compared to the prognostic models, which may reflect their reliance on the same inundation map [(Lines 409–413)].

I. 429: can you explain how the room for improvement can be compared between diagnostic and prognostic models?

Response: We have modified this paragraph. In our evaluation, prognostic models usually show lower R2 values or higher RMSE compared to the diagnostic models. This discrepancy likely reflects compounded uncertainty from internally simulated inundation and process complexity in the prognostic setups. As a result, improving the inundation product could be a key to improve these prognostic models [(Lines 463–467)].

**References**

- Chan, E., Worthy, D. E. J., Chan, D., Ishizawa, M., Moran, M. D., Delcloo, A., and Vogel, F.: Eight-Year Estimates of Methane Emissions from Oil and Gas Operations in Western Canada Are Nearly Twice Those Reported in Inventories, Environmental Science & Technology, 54, 14 899–14 909, https://doi.org/10.1021/acs.est.0c04117, 2020.
- Chang, R. Y.-W., Miller, C. E., Dinardo, S. J., Karion, A., Sweeney, C., Daube, B. C., Henderson, J. M., Mountain, M. E., Eluszkiewicz, J., Miller, J. B., Bruhwiler, L. M. P., and Wofsy, S. C.: Methane emissions from Alaska in 2012 from CARVE airborne observations, Proceedings of the National Academy of Sciences, 111, 16 694–16 699, https://doi.org/10.1073/pnas.1412953111, 2014.
- Granier, C., Darras, S., van der Gon, H. D., Doubalova, J., Elguindi, N., Galle, B., Gauss, M., Guevara, M., Jalkanen, J.-P., Kuenen, J., Liousse, C., Quack, B., Simpson, D., and Sindelarova, K.: The Copernicus Atmosphere Monitoring Service global and regional emissions (April 2019 version), https://atmosphere.copernicus.eu/sites/default/files/2019-06/cams\_emissions\_general\_document\_apr2019\_v7.pdf, copernicus Atmosphere Monitoring Service (CAMS) report, 2019.
- Hartery, S., Commane, R., Lindaas, J., Sweeney, C., Henderson, J., Mountain, M., Steiner, N., McDonald, K., Dinardo, S. J., Miller, C. E., Wofsy, S. C., and Chang, R. Y.-W.: Estimating regional-scale methane flux and budgets using CARVE aircraft measurements over Alaska, Atmospheric Chemistry and Physics, 18, 185–202, https://doi.org/10.5194/acp-18-185-2018, 2018.
- Hopcroft, P. O., Valdes, P. J., Wania, R., and Beerling, D. J.: Limited response of peatland CH4 emissions to abrupt Atlantic Ocean circulation changes in glacial climates, Climate of the Past, 10, 137–154, https://doi.org/10.5194/cp-10-137-2014, 2014.
- Ishizawa, M., Chan, D., Worthy, D., Chan, E., Vogel, F., Melton, J. R., and Arora, V. K.:
  Estimation of Canada's methane emissions: inverse modelling analysis using the
  Environment and Climate Change Canada (ECCC) measurement network,
  Atmospheric Chemistry and Physics, 24, 10 013–10 038,
  https://doi.org/10.5194/acp-24-10013-2024, 2024.
- Maasakkers, J. D., Jacob, D. J., Sulprizio, M. P., Turner, A. J., Weitz, M., Wirth, T., Hight, C., DeFigueiredo, M., Desai, M., Schmeltz, R., Hockstad, L., Bloom, A. A., Bowman, K. W., Jeong, S., and Fischer, M. L.: Gridded national inventory of U.S. methane emissions, Environmental Science & Technology, 50, 13 123–13 133, https://doi.org/10.1021/acs.est.6b02878, 2016.

- Maasakkers, J. D., McDuffie, E. E., Sulprizio, M. P., Chen, C., Schultz, M., Brunelle, L., Thrush, R., Steller, J., Sherry, C., Jacob, D. J., Jeong, S., Irving, B., and Weitz, M.: A gridded inventory of annual 2012–2018 U.S. anthropogenic methane emissions, Environmental Science & Technology, 57, 16 276–16 288, https://doi.org/10.1021/acs.est.3c05138, pMID: 37857355, 2023.
- MacKay, K., Lavoie, M., Bourlon, E., Atherton, E., O'Connell, E., Baillie, J., Fougère, C., and Risk, D.: Methane emissions from upstream oil and gas production in Canada are underestimated, Scientific Reports, 11, 8041, https://doi.org/10.1038/s41598-021-87610-3, 2021.
- Miller, S. M., Wofsy, S. C., Michalak, A. M., Kort, E. A., Andrews, A. E., Biraud, S. C., Dlugokencky, E. J., Eluszkiewicz, J., Fischer, M. L., Janssens-Maenhout, G., Miller, B. R., Miller, J. B., Montzka, S. A., Nehrkorn, T., and Sweeney, C.: Anthropogenic emissions of methane in the United States, Proceedings of the National Academy of Sciences, 110, 20018–20022, https://doi.org/10.1073/pnas.1314392110, 2013.
- Miller, S. M., Worthy, D. E. J., Michalak, A. M., Wofsy, S. C., Kort, E. A., Havice, T. C., Andrews, A. E., Dlugokencky, E. J., Kaplan, J. O., Levi, P. J., Tian, H., and Zhang, B.: Observational constraints on the distribution, seasonality, and environmental predictors of North American boreal methane emissions, Global Biogeochemical Cycles, 28, 146–160, https://doi.org/https://doi.org/10.1002/2013GB004580, 2014.
- Miller, S. M., Miller, C. E., Commane, R., Chang, R. Y.-W., Dinardo, S. J., Henderson, J. M., Karion, A., Lindaas, J., Melton, J. R., Miller, J. B., Sweeney, C., Wofsy, S. C., and Michalak, A. M.: A multiyear estimate of methane fluxes in Alaska from CARVE atmospheric observations, Global Biogeochemical Cycles, 30, 1441–1453, https://doi.org/https://doi.org/10.1002/2016GB005419, 2016b.
- Monforti Ferrario, F., Crippa, M., Guizzardi, D., Muntean, M., Schaaf, E., Lo Vullo, E., Solazzo, E., Olivier, J., and Vignati, E.: EDGAR v6.0 Greenhouse Gas Emissions, http://data.europa.eu/89h/97a67d67-c62e-4826-b873-9d972c4f670b, eDGAR database, European Commission, 2021.
- Oh, Y., Bruhwiler, L., Lan, X., Basu, S., Schuldt, K., Thoning, K., Michel, S. E., Clark, R., Miller, J. B., Andrews, A., Sherwood, O., Etiope, G., Crippa, M., Liu, L., Zhuang, Q., Randerson, J., van der Werf, G., Aalto, T., Amendola, S., Andra, S. C., Andrade, M., Nguyen, N. A., Aoki, S., Apadula, F., Arifin, I. B., Arnold, S., Arshinov, M., Baier, B., Bergamaschi, P., Biermann, T., Biraud, S. C., Blanc, P.-E., Brailsford, G., Chen, H., Colomb, A., Couret, C., Cristofanelli, P., Cuevas, E., Chmura, L., Delmotte, M., Emmenegger, L., Esenzhanova, G., Fujita, R., Gatti, L., Guerette, E.-A., Haszpra, L., Heliasz, M., Hermansen, O., Holst, J., Dilorio, T., Jordan, A., Jennifer, M.-W., Karion, A., Kawasaki, T., Kazan, V., Keronen, P., Kim, S.-Y., Kneuer, T., Kominkova, K., Kozlova, E., Krummel, P., Kubistin, D., Labuschagne, C., Langenfelds, R., Laurent,

- O., Laurila, T., Lee, H., Lehner, I., Leuenberger, M., Lindauer, M., Lopez, M., Mahdi, R., Mammarella, I., Manca, G., Marek, M. V., Mazière, M. D., McKain, K., Meinhardt, F., Miller, C. E., Mölder, M., Moncrieff, J., Moosen, H., Moreno, C., Morimoto, S., Myhre, C. L., Nahas, A. C., Necki, J., Nichol, S., O'Doherty, S., Paramonova, N., Piacentino, S., Pichon, J. M., Plass-Dülmer, C., Ramonet, M., Ries, L., di Sarra, A. G., Sasakawa, M., Say, D., Schaefer, H., Scheeren, B., Schmidt, M., Schumacher, M., Sha, M. K., Shepson, P., Smale, D., Smith, P. D., Steinbacher, M., Sweeney, C., Takatsuji, S., Torres, G., Tørseth, K., Trisolino, P., Turnbull, J., Uhse, K., Umezawa, T., Vermeulen, A., Vimont, I., Vitkova, G., Wang, H.-J. R., Worthy, D., and Xueref-Remy, I.: CarbonTracker CH4 2023, https://doi.org/10.25925/40JT-QD67, 2023.
- Ringeval, B., Hopcroft, P. O., Valdes, P. J., Ciais, P., Ramstein, G., Dolman, A. J., and Kageyama, M.: Response of methane emissions from wetlands to the Last Glacial Maximum and an idealized Dansgaard–Oeschger climate event: insights from two models of different complexity, Climate of the Past, 9, 149–171, https://doi.org/10.5194/cp-9-149-2013, 2013.
- Scarpelli, T., Jacob, D., Moran, M., Reuland, F., and Gordon, D.: Gridded inventory of Canada's anthropogenic methane emissions for 2018, https://doi.org/10.7910/DVN/CC3KLO, 2021.
- Zhang, Z., Poulter, B., Melton, J. R., Riley, W. J., Allen, G. H., Beerling, D. J., Bousquet, P., Canadell, J. G., Fluet-Chouinard, E., Ciais, P., Gedney, N., Hopcroft, P. O., Ito, A., Jackson, R. B., Jain, A. K., Jensen, K., Joos, F., Kleinen, T., Knox, S. H., Li, T., Li, X., Liu, X., McDonald, K., McNicol, G., Miller, P. A., Müller, J., Patra, P. K., Peng, C., Peng, S., Qin, Z., Riggs, R. M., Saunois, M., Sun, Q., Tian, H., Xu, X., Yao, Y., Xi, Y., Zhang, W., Zhu, Q., Zhu, Q., and Zhuang, Q.: Ensemble estimates of global wetland methane emissions over 2000–2020, Biogeosciences, 22, 305–321, https://doi.org/10.5194/bg-22-305-2025, 2025.